# Nuclear and cytoplasmic specific RNA binding proteome enrichment and its changes upon ferroptosis induction

Haofan Sun [1], Bin Fu[1], Xiaohong Qian[1], Ping Xu [1] & Weijie Qin [1,2] ✉

The key role of RNA-binding proteins (RBPs) in posttranscriptional regulation of gene expression is intimately tied to their subcellular localization. Here, we show a subcellular-specific RNA labeling method for efficient enrichment and deep profiling of nuclear and cytoplasmic RBPs. A total of 1221 nuclear RBPs and 1333 cytoplasmic RBPs were enriched and identified using nuclear/cytoplasm targeting enrichment probes, representing an increase of 54.4% and 85.7% compared with previous reports. The probes were further applied in the omics-level investigation of subcellular-specific RBP-RNA interactions upon ferroptosis induction. Interestingly, large-scale RBPs display enhanced interaction with RNAs in nucleus but reduced association with RNAs in cytoplasm during ferroptosis process. Furthermore, we discovered dozens of nucleoplasmic translocation candidate RBPs upon ferroptosis induction and validated representative ones by immunofluorescence imaging. The enrichment of Tricarboxylic acid cycle in the translocation candidate RBPs may provide insights for investigating their possible roles in ferroptosis induced metabolism dysregulation.

RNA-binding proteins (RBPs) are key components that regulate gene expression in eukaryotes[1]. The interactions between RBPs and RNAs play a vital role in modulating the maturation, stability, localization, editing and translation of RNA transcripts[2,3]. Dysfunctional RBPs have been associated with various human diseases, such as neurodegeneration, autoimmune disease, and cancers[4,5]. Therefore, extensive efforts have been made to systematically investigate RBPs, not only to gain a better understanding of their complex interplay with RNAs in maintaining normal physiological function, but also to gain insight into how their dysregulation functions in many diseases[6]. The roles of biomolecules are usually regulated by their presence in specific organelle, in which they meet with different partners under different environment[7–9]. The function of RBPs is also subjected to their subcellular localizations because posttranscriptional regulation usually occurs in the specific subcellular compartments that are formed by membrane separation and phase separation[10]. Different compartments provide different chemical environments and various potential interaction partners or substrates for RBPs. Furthermore, RBPs control the subcellular localization of RNA substrates by binding to motor proteins or anchoring proteins and suppress mRNA translation before the destination location is reached[11]. Tight regulation of RBP subcellular localization can be considered as an important layer of control over cell physiology. Therefore, comprehensively identifying RBPs from specific subcellular compartments and quantifying their dynamic changes under cell stress are of great importance for elucidating their regulatory roles in biological functions and pathological processes[12,13]. Systematically mapping the subcellular localized RBPs and their dynamic assembly in ribonucleoprotein complexes (RNPs) requires highly efficient methods to achieve specific and unbiased enrichment of RBPs in particular organelles.

For global RBP and RNP enrichment, 254 nm ultraviolet (UV) crosslinking is a classic strategy that generates covalent bonds between RBPs and RNA in living cells[14]. Large-scale analysis of RBPs can be achieved by the UV crosslinking and immunoprecipitation (CLIP)

[1]State Key Laboratory of Medical Proteomics, Beijing Proteome Research Center, National Center for Protein Sciences (Beijing), Beijing Institute of Lifeomics, Beijing 102206, China. [2]College of Chemistry and Materials Science, Hebei University, Baoding 071002, China. ✉e-mail: aunp_dna@126.com

method and its derivatives[15–19]. Although the subcellular locations of RBPs can be obtained from the UniProt database, many RBPs exist in multiple subcellular sites but tend to bind to RNA in only one of these locations, such as the mRNA-assistant nucleocytoplasmic transport of RBPs[20–22]. Therefore, high-throughput RBP profiling methods with organelle spatial resolution are of great significance for understanding the diverse roles of RBPs in different subcellular compartments. Recently, a few hundred nuclear RBPs were identified by mass spectrometry (MS) analysis using enrichment methods such as serial RNA interactome capture (SerIC), RNA-binding region identification (RBR-ID) and APEX-mediated proximity biotinylation with organic-aqueous phase separation (OAPS)[23–25]. These methods promoted the mapping of nuclear RBPs and lead to the discovery of new functions of nuclear RBPs in regulating cellular pathways. Despite these pioneer works in the nuclear RNA-binding proteome, current methods still face significant technical challenges. Serial fractionation to obtain the organelles only provides limited purity, which may not be satisfactory for subcellular RBP analysis[26]. APEX-PS requires the cells to be genetically manipulated, which is difficult to achieve with hard-to-transfect cells (e.g., macrophage cells), primary neurons, or tissue samples[27]. Furthermore, the complex multiple-step procedures for serially enriching the nucleus (proteome) and nuclear RBPs in the above methods may result in severe sample loss and limit the broad application of these methods. Therefore, new methods without these limitations are urgently needed to facilitate the identification of RBPs that are located in specific organelles and the discovery of their subcellular compartment-related functions.

In this work, to address the above issues, subcellular-specific RNA labeling probes are developed to achieve efficient enrichment and deep profiling of nuclear and cytoplasmic RNPs/RBPs (Fig. 1a). These probes are composed of the following parts: the furocoumarin part for RNA tagging, the subcellular targeting part using nucleus and cytoplasm localizing peptides and a biotin part as a handle for RNPs/RBP enrichment (Fig. 1a). Furocoumarins are a class of planar, three-ring heterocyclic compounds that can form covalent bonds via cycloaddition reactions with uridine in RNA upon irradiation with 320–380 nm UV light[28,29]. Therefore, efficient light-induced RNA tagging and comprehensive isolation of both coding and noncoding RNPs can be achieved via cycloaddition reactions between furocoumarin and uracil. For nuclear and cytoplasmic localization of the probe, transactivating transcriptional activator (TAT), nuclear localization signal (NLS) and nuclear export signal (NES) peptides were chosen in this work. The TAT peptide has the ability to enter various cell types while maintaining the integrity of the cell membrane[30]. The TAT peptides were applied to improve the efficiency of intracellular delivery of the probe[31,32]. An NLS peptide is an amino acid sequence that tags proteins to import them into the cell nucleus by nuclear transport, while an NES peptide tags proteins to export them from the cell nucleus to the cytoplasm[33,34]. The combination of TAT-NLS/TAT-NES with furocoumarin in this work provides the probes with an ability to perform subcellular targeting for specific enrichment of nuclear and cytoplasmic RBPs. Organelle-specific tagging using TAT-NLS-furocoumarin and TAT-NES-furocoumarin was demonstrated by confocal fluorescence imaging. RNP isolation and RBP enrichment from HeLa cells using the biotin handle in the probes led to the identification of 1221 nuclear RBPs and 1333 cytoplasmic RBPs by MS analysis and 54.4% and 85.7% enhancement compared with previous reports[23,35]. Subcellular localization analysis of the identified RBPs further supported their organelle distribution. A total of 83.0% of RBPs enriched by TAT-NLS-furocoumarin are located in the nucleus, and 95.3% of RBPs enriched by TAT-NES-furocoumarin are located in the cytoplasm. The probes were successfully applied to investigate the changes of the RNPs in the nucleus and cytoplasm upon ferroptosis induction. Translation-related RBPs exhibit enhanced interactions with RNAs in the nucleus but decreased binding with RNAs in the cytoplasm after a treatment with ferroptosis

inducers, indicating ferroptosis induction could disturb protein translation. Furthermore, a number of nucleoplasmic translocation candidate RBPs involved in multiple pathways, such as cellular response to stress/stumli, DNA synthesis and replication and TCA cycle were found upon different ferroptosis inducer treatment, which may provide new insights for investigating their roles in ferroptosis induced pathway dysregulation. The above finding suggests the potential of our organelle-specific RNA tagging probes as a high-throughput method to perform spatially resolved mapping of RBPs upon cellular perturbation.

## Results

### Establishment of the subcellular RNP enrichment strategy

In this work, we developed subcellular-specific RNP labeling probes for efficient enrichment and deep profiling of nuclear and cytoplasmic RBPs (Fig. 1a). First, the RNA tagging probes were incubated with living cells in the culture dish to allow the probes to enter the cells and to position to the nucleus or cytoplasm regions. Next, 254 nm and 365 nm UV irradiation was applied to the cells to crosslink RNA and RBPs and to tag the probe on the RNA. After cell lysis, streptavidin beads were applied to the cell lysate to enrich the crosslinked RNPs via the biotin handle on the probes. After several washings were performed to remove the nonspecifically adsorbed non-RBPs, the enriched RNPs were released from the beads and subjected to either RNA sequencing or mass spectrometry (MS) analysis. As the key component of this organelle-targeting RNA tagging method, the subcellular-specific probes were composed of the following parts: an uracil reactive furocoumarin group for RNA tagging, subcellular targeting peptides for nucleus/cytoplasm localization and a biotin handle for enrichment (Fig. 1a). Different combinations of TAT, NLS and NES units and the corresponding synthetic routes were used for preparing the trifunctional probes (Table 1 and Fig. S1, Supporting Information). The MS spectra and molecular weight of the probes confirm successful synthesis (Fig. S2 and Table S1, Supporting Information).

We first validated the ability of these probes to be located in the designated subcellular locations. Confocal fluorescence images using FITC-streptavidin labeling via the biotin handle of the probes showed that BLF, BLTF and BL3F were correctly localized in the nucleus and BETF, BELF and BLEF were in the cytoplasm, indicating that these probes successfully targeted their respective organelles (Fig. 1b, c). Interestingly, simultaneously applying NLS and NES in the same probe (BELF and BLEF) resulted in complete probe localization in the cytoplasm, which may be attributed to the signal strength of NLS was weaker than that of NES. BTF could enter the cells but failed to target specific organelles. The probe was distributed throughout the whole cell. BEF could not enter the cells, which may be attributed to the absence of TAT peptides or NLS in the probe and the lack of a membrane-penetrating ability. Therefore, these two probes were excluded from further analysis. The above results demonstrated that either the TAT peptide or NLS is necessary for efficient intracellular localization of the probe; therefore, probes without the TAT peptide or NLS were excluded from further analysis. Next, the RNP enrichment capability of each of the remaining probes was verified by SDS–PAGE after the bead-captured RBPs were eluted by a RNase A treatment (Fig. S3, Supporting Information). The RNase A treatment decomposed the RNAs in the RNPs and led to the release of the RBPs, which were visualized in the gel image after silver staining. For all six probes that were tested, clear protein bands of the probe enrichment products were observed in the gel. In contrast, protein bands were barely observed without probe or 254/365 nm UV treatment, indicating the high selectivity of this enrichment method with only marginal non-specific adsorption of non-RBPs. Using RNase A to wash the beads resulted in almost completely abolished protein bands, confirming the RNA dependence of the observed protein bands (Fig. 1d). We further

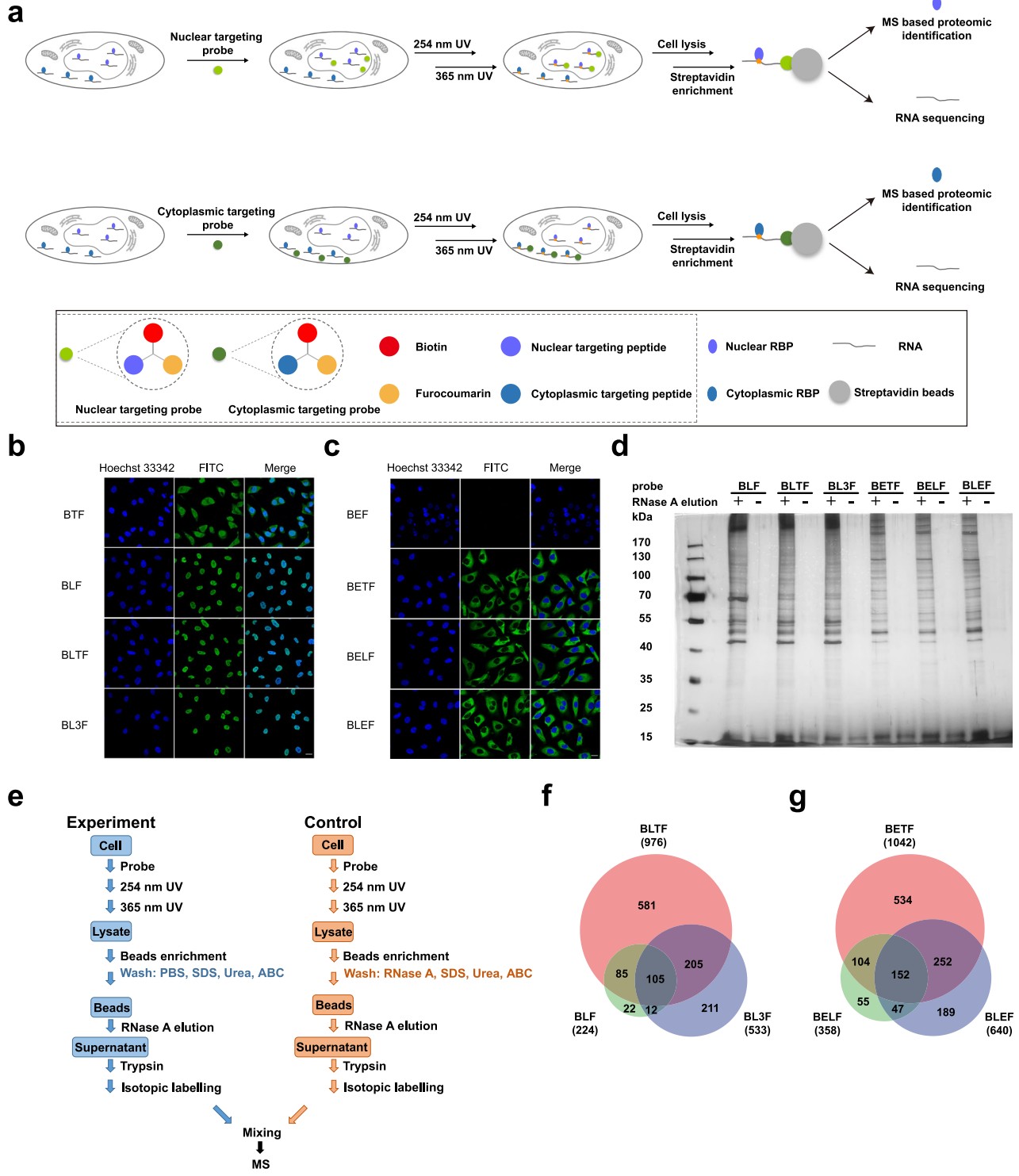

**Fig. 1 | Subcellular-specific RNA tagging and RBP enrichment by furocoumarin probes. a** Schematic overview of subcellular-specific RNA tagging and RBP enrichment for RNA sequencing and MS-based RBP identification. Confocal fluorescent images of (**b**) nucleus-targeting probes localization and (**c**) cytoplasm-targeting probes localization. The probes were visualized by FITC (green), and Hoechst 33342 (blue) was used as a nuclear marker. Scale bars, 20 µm. **d** SDS-PAGE characterization of probe enriched RBPs with and without RNase A washing. **e** Experimental design of the quantitative differential proteomic comparison between the experimental group and control group for RBP identification. Overlap of the RBPs identified by (**f**) BLF, BLTF and BL3F and by (**g**) BETF, BELF and BLEF. Source data are provided as a Source Data file.

evaluated the most appropriate concentration of each probe for RBP enrichment. Generally, 1–5 µM probes resulted in the most intensive RBP bands in the gel (Fig. S4, Supporting Information); therefore, 2 µM probes were chosen to incubate the cells for organelle-specific RNA tagging in the subsequent tests.

## Proteomic identification and RNA sequencing of the subcellular-specific RNPs enriched by organelle-target probes

For large-scale MS-based subcellular RBP profiling, the key step for obtaining reliable RBPs was to distinguish and remove the non-RBPs from the results. Therefore, RBPs were identified using a differential

proteomics strategy by quantitatively comparing the experimental and control groups. The only difference between the experimental and control groups was the addition (control) or lack of addition (experiment) of RNase A in the washing step (Fig. 1e). Washing the streptavidin beads with RNase A in the control group resulted in RNA degradation and removed the RNPs from the sample. Therefore, the subsequent elution with RNase A could not produce RBPs in the control group. In contrast, RNase A was only applied in the elution step for the experimental group. In this way, specific elution of the bead-enriched RBPs into the supernatant was achieved for subsequent MS analysis without possible coelution of DNA-binding proteins. Stable isotopic dimethyl labeling was applied to combine the experimental group and control group for differential proteome analysis to screen the true RBPs. The detailed flow chart further explained the exact processing procedures and conditions used for the "experiment" samples and the controls (Fig. S5). Quantitative comparison between the experimental and control groups resulted in the identification of 224, 976 and 533 highly confident nuclear RBPs by BLF, BLTF and BL3F enrichment using stringent screening criteria that was reported in previous works (fold change >2, $P < 0.01$ and FDR <1% with two or greater unique peptides in at least two tests[19,36,37], Fig. S6a–c, Supplementary Data 1). For cytoplasmic RBP identification, 1042, 358 and 640 highly confident cytoplasmic RBPs were identified by BETF, BELF and BLEF enrichment using the same screening criteria (Fig. S6d–f, Supplementary Data 1). Good reproducibility was achieved for all six probes, with Pearson coefficients ranging from 0.79 to 0.96 (Fig. S7, Supporting Information). In total, 1221 nuclear RBPs and 1333 cytoplasmic RBPs were identified by the nucleus-targeting probes and the cytoplasm-targeting probes (Supplementary Data 1, Supporting Information). Overlapping but distinct RBPs were identified by different nucleus/cytoplasm-targeting probes (Fig. 1f, g). BLTF covered most of the identified nuclear RBPs (79.9%) and BETF covered the majority of the identified cytoplasmic RBPs (78.2%). The above results indicated the advantage of the combined application of TAT and NLS/NES for both plasma membrane penetration and nucleus/cytoplasm localization of the probes, which was not reported in previous works. The RBPs that were enriched by different probes exhibited a high overlap with a compiled list of 5306 human RBPs, which has been previously annotated by RBP profiling methods[19,36–43]. A total of 98.2%, 97.1% and 95.5% of the RBPs obtained by the nucleus-targeting probes (BLF, BLTF and BL3F) and 96.2%, 95.5% and 96.1% of the RBPs obtained by the cytoplasm-targeting probes (BETF, BELF and BLEF) were previously reported as RBPs.

Next, we conducted pathway and gene ontology (GO) analysis using all the identified RBPs that were obtained by different probes. The high RNA correlation of the identified RBPs was confirmed by the top enriched pathways, which were almost all associated with RNA processing, such as rRNA processing, translational silencing, mRNA splicing and translation termination (Fig. 2a). Consistently, gene ontology (GO) analysis revealed highly RNA-related GO terms, including translation, rRNA processing, and mRNA splicing for biological processes and RNA binding, mRNA binding and translation regulator activity for molecular functions, indicating that the known RBPs were overrepresented in our dataset (Fig. 2b–e). Gene Ontology cellular component (GOCC) analysis showed that nucleus-related terms, such as nucleoplasm and nucleolus, were significantly enriched in the RBPs obtained by the nucleus-targeting probes. Further comparison with the UniProt database revealed that 78.8% of RBPs obtained by the nucleus-targeting probes were located in the nucleus. Similarly, GOCC analysis exhibited high cytoplasm-related terms, such as cytoplasm and cytosol, for the RBPs obtained by the cytoplasm targeting probes. An even higher specificity was obtained for the cytoplasm-targeting probes, with 91.1% RBPs located in the cytoplasm by comparing with the UniProt database. Considering their efficient enrichment of RBPs with high subcellular targeting capability, BLTF and BETF were

**Table 1 | Detailed design of the probes**

| Abbreviation | Target | Constitution |
|---|---|---|
| BTF | nucleus | Biotin-TAT-Furocoumarin |
| BLF | nucleus | Biotin-NLS-Furocoumarin |
| BLTF | nucleus | Biotin-NLS-TAT-Furocoumarin |
| BL3F | nucleus | Biotin-NLS-NLS-NLS-Furocoumarin |
| BEF | cytoplasm | Biotin-NES-Furocoumarin |
| BETF | cytoplasm | Biotin-NES-TAT-Furocoumarin |
| BELF | cytoplasm | Biotin-NES-NLS-Furocoumarin |
| BLEF | cytoplasm | Biotin-NLS-NES-Furocoumarin |

The corresponding synthetic routes are shown in Fig. S1.

selected as the optimized probes for nuclear and cytoplasmic RBP enrichment in the following experiments.

To experimentally investigate the type and distribution of the subcellular targeting probes isolated RNAs, proteinase K was applied to the probe enriched RNPs to digest RBPs and remove them from the RNAs. Control experiments were also conducted by first extracting the nucleus and cytoplasm using density gradient centrifugation and then separately isolating the nuclear and cytoplasmic RNAs via the traditional TRIzol method. Intronic regions are enriched in the nuclear RNA isolated by BLTF compared with the cytoplasmic RNA obtained by BETF (Fig. S8, Supporting Information). Similar trends were found in the control experiments. Furthermore, intergenic regions accounts for only about 5% and 1% of reads in the enrichment result of BLTF and BETF, indicating relatively low levels of DNA contamination[44–46]. Consistent RNA distribution patterns between the BLTF isolated RNAs and the control nuclear RNAs were found by RNA sequencing (Fig. 3a–d). Apart from rRNA, which was removed from the extracted total RNA using a SEQuoia RiboDepletion kit[47], wide variety types of RNA were obtained by BLTF enrichment, including protein coding RNA and noncoding RNAs, such as lincRNA, antisense RNA, miRNA and snoRNA. Similarly, the BETF-isolated RNAs and control cytoplasmic RNAs exhibited consistent distribution patterns. Various types of RNA were enriched by BETF, including mRNA, lincRNA, miRNA and antisense RNA. The isolated RNA samples were assessed by Agilent 2100 Bioanalyzer before ribodepletion. Clear, sharp bands of 18S rRNA and 28S rRNA with minimal RNA degradation were observed (Fig. 3e, f). The above results demonstrated that BLTF and BETF were capable of enriching RNPs that contain various kinds of RNAs.

**Comparison of the identified nuclear RBPs to cytoplasmic RBPs**
Domain analysis of the RBPs identified by BLTF and BETF showed that classical RNA binding domains (RBDs), such as RRM, DEAD and KH, as well as nonclassical RBDs were among the top represented domains. No particular RBD differences were found between the two organelle-specific RBPs (Fig. 4a). The physiochemical characteristics of the RBPs identified by the two probes were compared, including their positively charged amino acids, isoelectric points, hydrophobicity and disorder (Fig. 4b–e). Similar distribution patterns were found for the two probes, but BLTF-enriched nuclear RBPs displayed a slightly higher positively charged amino acid, isoelectric point, and disorder. To further investigate the RBPs enriched by the nucleus-targeting BLTF and cytoplasm-targeting BETF, a volcano plot was prepared to illustrate the difference between the two groups of RBPs (Supplementary Data 2, Supporting Information). Only the identified RBPs that passed the screening cut-off (red dots in Fig. S6) were used (Fig. 4f). Only a small fraction of RBPs were co-identified by both probes, and this result may be attributed to the colocalization of RBPs in both nucleus and cytoplasm (Fig. 4f). The remaining 79.9% of RBPs were primarily enriched by only one of the probes, indicating their major subcellular location for RNA binding. The above results also verified the ability of BLTF and

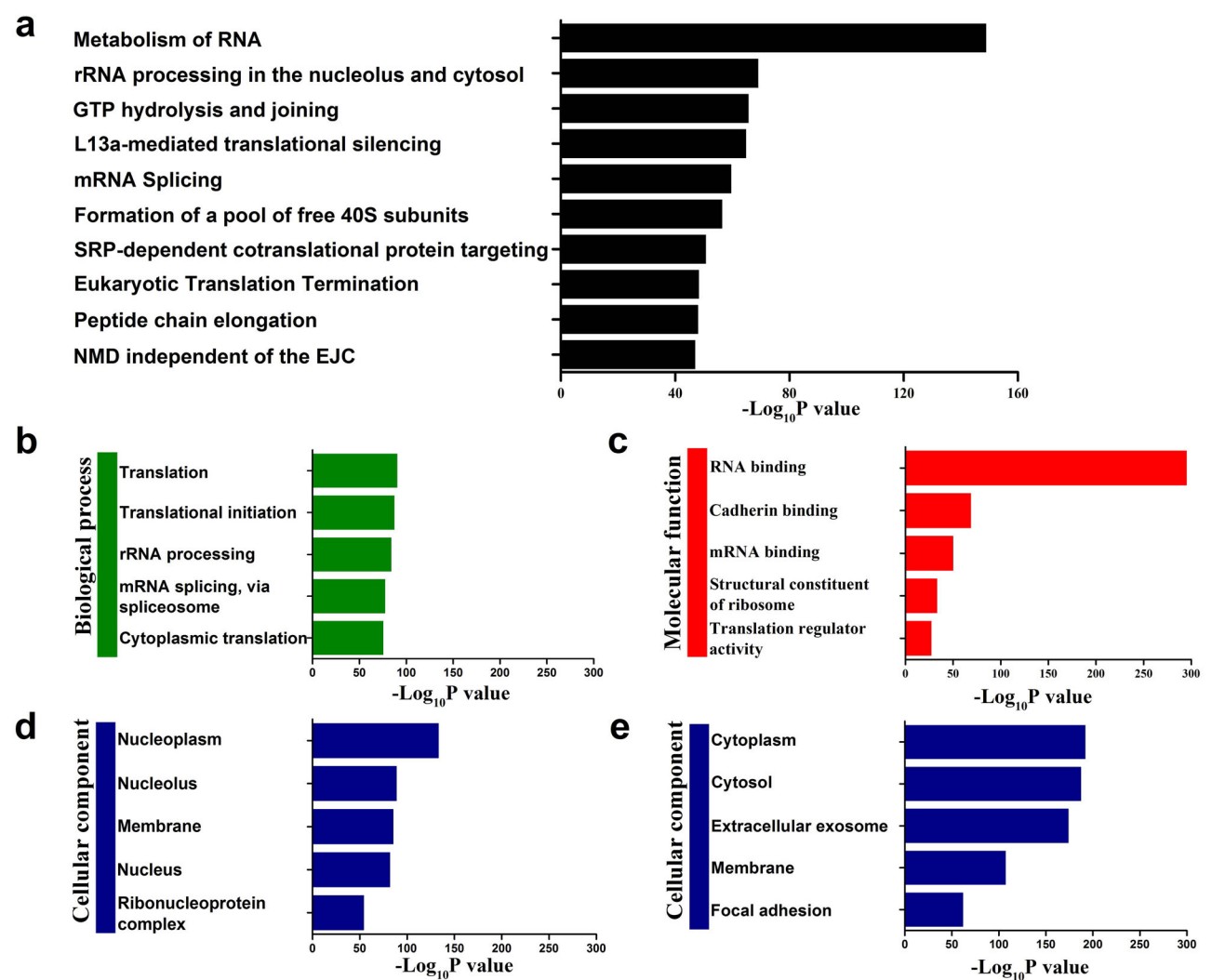

**Fig. 2 | Pathway and GO analysis of the enriched RBPs. a** Pathway, (**b**) GO biological process and (**c**) GO molecular function of the RBPs obtained by all probes. GO cellular components of the RBPs obtained by (**d**) nucleus-targeting probes and (**e**) cytoplasm-targeting probes. Statistical analysis was performed with hypergeometric test (Benjamini&Hochberg (BH) adjusted *P* values). Source data are provided as a Source Data file.

BETF to locate and enrich RBPs in distinct subcellular compartments, which is crucial for studying the subcellular localization-related functions of RBPs and their dynamic changes upon external perturbations.

Next, how RNA binding quantity (RBQ) of the RBPs correlates with their abundance in each compartment was determined by subcellular RBP enrichment and organelle specific proteome abundance profiling. Nucleus and cytoplasm fractions were obtained by density gradient centrifugation using the protocol described by Conrad et al.[24] for proteome profiling. The purified nucleus and cytoplasm fractions were evaluated by immunoblotting using nuclear markers lamin A/C and histone 3, as well as cytoplasmic marker β-tubulin (Fig. S9, Supporting Information). The nuclear fraction only shows clear bands of Lamin A/C and histone 3, while the cytoplasmic fraction shows β-tubulin band without contamination of nuclear markers, demonstrating excellent nuclear and cytoplasmic purification results. The scatter plot showed how the RBQ correlates with the RBP abundance in nucleus and cytosol (Fig. 5a). Most of the RBPs are located in the first and third quadrants indicating that the subcellular-level RBQ of majority of RBPs is consistent with their abundance distribution in the corresponding organelle. We further analyzed the rest RBPs that fall in the second and fourth quadrants, which have opposite behavior in organelle specific RBQ and RBP abundance distribution (Fig. 5a). GO analysis showed the RBPs with obviously higher abundance but relatively lower RBQ in

cytoplasm (Fig. 5b). RBPs with GO biological process terms such as 'DNA replication', 'regulation of mRNA stability' and 'DNA replication initiation' are enriched. For RBPs with opposite behavior and enhanced abundance but decreased RBQ in nucleus, GO biological process terms such as 'cytoplasmic translation', 'rRNA processing', and 'translational initiation' are enriched.

## Dysregulated subcellular-specific RNA-RBP interactions in ferroptosis

Ferroptosis is a recently recognized form of programmed cell death which involves overwhelming iron-dependent accumulation of cellular reactive oxygen species (ROS)[48,49]. It plays important regulatory roles in a broad set of biological contexts and diseases, such as tumors, acute organ injury, and neurodegeneration[50–52]. Different RBPs, such as GPX4, RBMS1 or ZFP36/TTP that promote or protect cells against ferroptosis were reported, indicating the complex role of RBPs as essential regulators in ferroptosis[53–55]. Furthermore, ferroptosis was reported to be associated with nucleocytoplasmic transport of proteins, which is intimately tied to RBP functions, as post-transcriptional regulations are often carried out in subcellular compartments[10,56–58]. However, omics-level investigation on alternation of the RBP-RNA interaction and nucleocytoplasmic distribution upon ROS accumulation during the ferroptosis process was not reported. Therefore, three

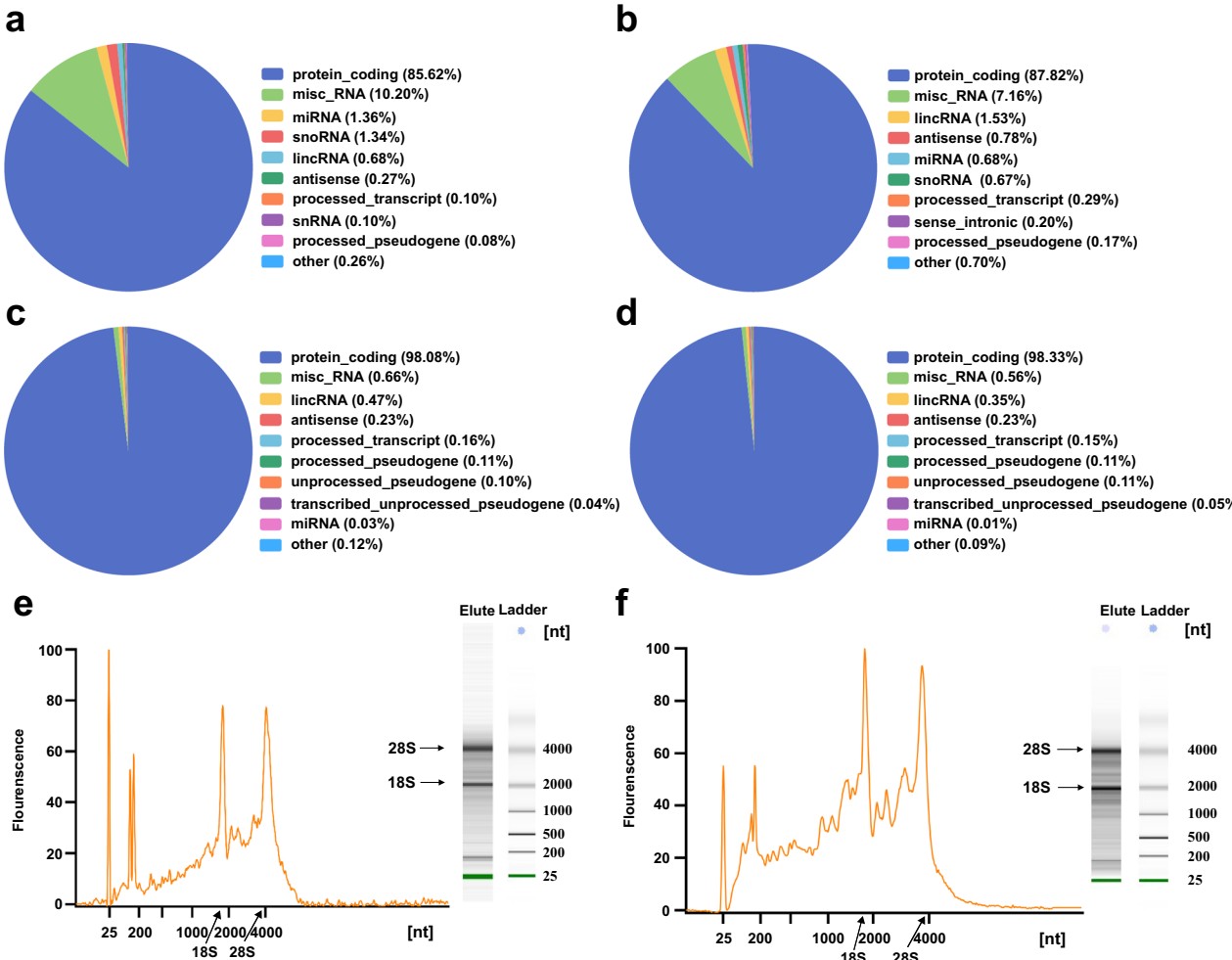

**Fig. 3 | Pie chart and capillary electrophoresis analyzing of the type and distribution of the subcellular specific RNAs.** Nuclear RNAs isolated by (**a**) density gradient centrifugation and (**b**) BLTF. Cytoplasmic RNAs isolated by (**c**) density gradient centrifugation and (**d**) BETF. Reads for each GENCODE biotype were normalized to the total number of reads in one library. Capillary electrophoresis analyzing of the (**e**) BLTF and (**f**) BETF enriched RNAs (three biological replicates). Source data are provided as a Source Data file.

ferroptosis inducers, erastin from voltage-dependent anion channel receptor pathway and system Xc⁻ pathway[59,60], RSL3 from glutathione/glutathione peroxidase 4 (GSH/GPX4) pathway[53] and DAT from iron metabolism pathway[61] were applied to determine changes of the RNA-RBP interaction and variation of the subcellular localization of RBPs in ferroptosis. In vitro biocompatibility of these ferroptosis inducers were investigated by examining the cell survival rates (Fig. S10a–c, Supporting Information). No significant cytotoxicity was found after 2 h treatment with 20 µM erastin, 4 h treatment with 0.5 µM RSL3, or 4 h treatment with 50 µM DAT. Less than 6% of the cells were double positive for Annexin-V and PI staining, indicating that the cells were still in early stage of ferroptosis with no noticeable cell death[62] (Fig. S10d, Supporting Information). Confocal fluorescent imaging analysis was performed to show the localization of the BLTF/BETF probes before and after ferroptosis treatment (Fig. S11, Supporting Information). BLTF/BETF probes do not alter their originally targeted subcellular localization after ferroptosis treatment. Furthermore, MitoTracker and Hoechst 33342 staining indicated that integrity of mitochondria and nucleus was not affected by the ferroptosis inducers.

We then investigated the influence of early ferroptosis on subcellular-specific proteome abundance and RNA-RBP interactions. Only very small fractions (2.1–9.3%) of the nuclear/cytoplasmic proteome shows significant up- or downregulation, indicating marginal variation in early ferroptosis (Fig. S12, Supplementary Data 3). RNPs in the ferroptosis inducer-treated and nontreated cells were enriched using nucleus/cytoplasm targeting BLTF/BETF probes. Quantitative proteomics analysis revealed a clear trend of RBPs upregulation in nuclear RNPs and downregulation in cytoplasmic RNPs after exposure to erastin with a greater extent perturbation of cytoplasmic RBPs (Fig. S13, Supporting Information). For example, the binding of 238 nuclear RBPs with RNAs were markedly increased and six were decreased, while only one cytoplasmic RBPs displayed notably increased association with RNAs but 377 exhibited decreased association (Fig. 6a, Supplementary Data 4). Similarly, large scale downregulation of RBPs was also found in cytoplasmic RNPs in the RSL3 treated cells. Perturbation of the organelle-specific RBP-RNA interaction was linked with modulation of several GO biological processes. The RBPs with upregulated RNA-binding in nucleus and downregulated RNA-binding in cytoplasm were highly related to the GO term 'translation' in Erastin and RSL3 treated cells, indicating translation-related RBPs were accumulated in nucleus and/or were disassociated with corresponding RNAs in cytoplasm upon ferroptosis induction (Fig. 6b). A closer investigation revealed moderate overlap of the downregulated RBPs in cytoplasm RNPs and relatively low overlap of the upregulated RBPs in nuclear RNPs among erastin, RSL3 and DAT treated cells, which indicates the three inducers may perturb nucleus and cytoplasm via different pathways (Fig. S14a, b, Supporting

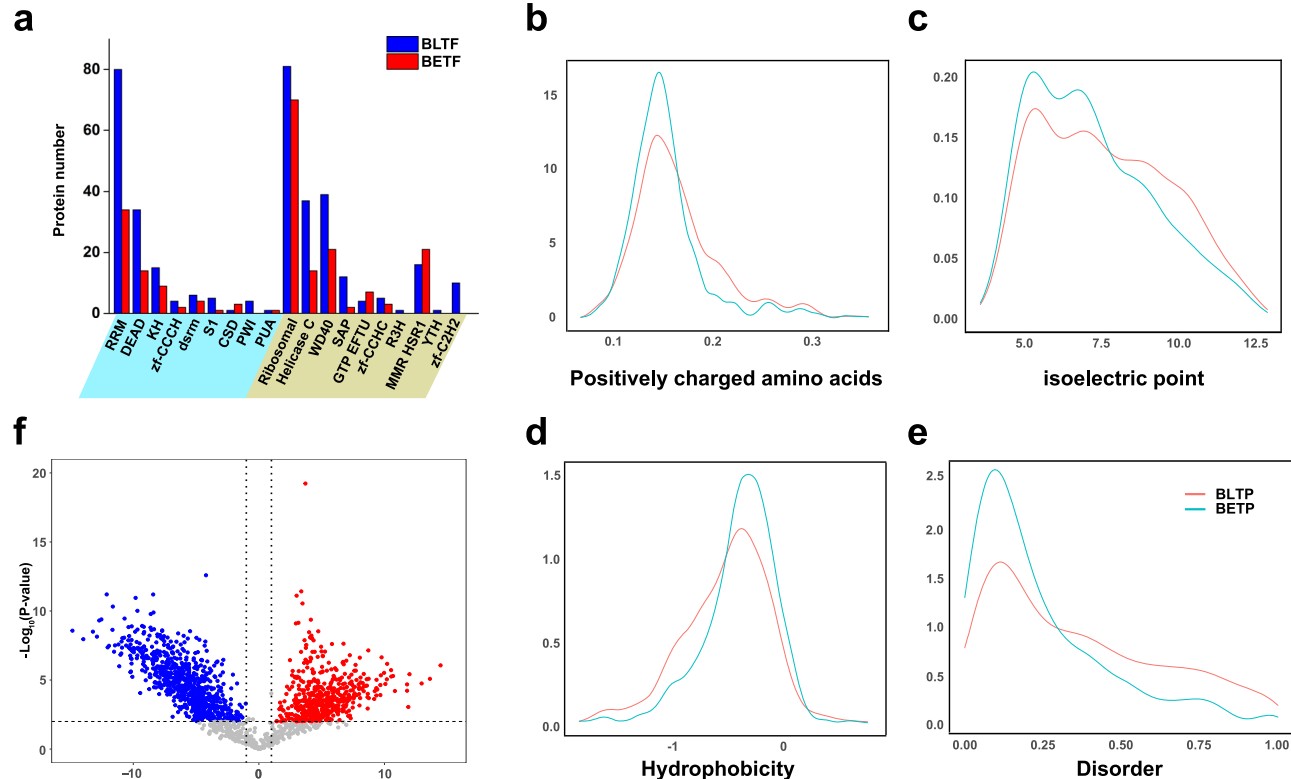

**Fig. 4 | Comparison of the identified nuclear RBPs to cytoplasmic RBPs.**
**a** Number of the identified RBPs with classical (left) and non-classical RBDs (right).
BLTF identified RBPs were depicted in blue and BETF ones were in red. **b** Percentage
of positively charged amino acids, (**c**) distribution pattern of isoelectric point, (**d**)
hydrophobicity and (**e**) disorder of the BLTF identified nuclear RBPs (red line) and
BETF identified cytoplasmic RBPs (cyan line). **f** Volcano plot displaying the $\log_2$ fold
change (FC) (x-axis) and −log $P$ values (y-axis) of the RBPs identified by BLTF and
BETF. The red dots represent RBPs in the nucleus, and the blue dots represent RBPs
in the cytoplasm. Statistical analysis was performed with two-sided Student's $t$ test
(BH adjusted $P$ values) from three biological replicates. Source data are provided as
a Source Data file.

Information). The 59 RBPs displayed both nuclear upregulation and cytoplasmic downregulation by erastin treatment were considered as nucleoplasmic translocation candidates (Fig. S14c, Supplementary Data 4). More than half of these candidates were involved in the pathway term 'cellular response to stress/stumli' pathway (Fig. S15a, Supporting Information). DNA synthesis and replication was also enriched in the pathway and GO analysis, which was consistent with previous studies of erastin induced ferroptosis[63,64] (Fig. S 15a, b, Supporting Information). One third of these candidates were involved in 'Metabolism' pathway including Tricarboxylic acid (TCA) cycle (Fig. S 15a, b, Supporting Information). TCA cycle has long been related to RNA binding activities[65] and was also reported to promote cellular GSH deletion leading to ferroptosis by GSH/GPX4 pathway[66,67].

To visualize abundant changes of the RBPs in the nucleus and cytoplasm, immunofluorescence imaging was used to compare the samples with and without ferroptosis induction. RPL7A and RPS27A were selected as negative controls without obvious change in both subcellular proteome profiling and RNA binding enrichment after erastin treatment in both nucleus and cytoplasm (Table S2, Supporting Information). The immunofluorescence images further confirmed that the nuclear/cytoplasmic ratios of RPL7A and RPS27A were not changed after treatment with erastin (Fig. S16, Supporting Information). For the 24 translation-related RBPs exhibited enhanced interaction with RNAs in nucleus after erastin induced ferroptosis, only EIF5 displays upregulation in the nuclear proteome profiling dataset. To further validate the nucleoplasmic translocation of the translation-related RBPs, we chose one RBP as the representative for each of the three situations in our study. EIF5 represents upregulation in both RBP enrichment by the nuclear BLTF probe and proteome profiling in the nucleus fraction

after erastin treatment. FXR1 represents upregulation only in RBP enrichment, while the protein abundance remained unchanged in proteome profiling. Besides the above two situations, ECHS1 was also chosen as a negative control to represent RBPs that were unchanged in both RBP enrichment and proteome profiling. Immunofluorescence imaging displayed consistent results with the proteome profiling data of the nuclear fraction (Fig. 7). The results demonstrated that EIF5 translocated to nucleus with 72.7% fluorescence enhancement after erastin induction. FXR1 and ECHS1 showed no abundance change in the nucleus (Fig. 7d). Therefore, the increased FXR1 in RBP enrichment after erastin treatment should be attributed to the enhanced binding with the corresponding RNAs in the nucleus. Next, translocation of TCA proteins to the nucleus was validated using subcellular fractionation followed by proteomics analysis and fluorescent imaging. The five TCA proteins that were upregulated in nuclear RBP enrichment after erastin treatment were chosen for further analysis. ACO2 and FH were upregulated, while CS and MDH2 were unchanged after inducing ferroptosis (Fig. S17, Supporting Information). The remaining one (PDHA1) was not identified in the proteome of nuclear fraction. Consistent results were obtained using immunofluorescence imaging (Fig. S18, Supporting Information). ACO2 and FH were indeed translocated to the nucleus with 31.1% and 19.2% fluorescence enhancement after erastin treatment, while changes in nucleocytoplasmic distribution of CS, MDH2 and PDHA1 were negligible. We further normalized the quantitative changes of nuclear RBP enrichment by their corresponding nuclear protein abundance variation. All the four TCA related RBPs still displayed upregulated RNA binding after normalization (Fig. S19, Supporting Information). The above results revealed that both increased protein abundance by translocation to nucleus and

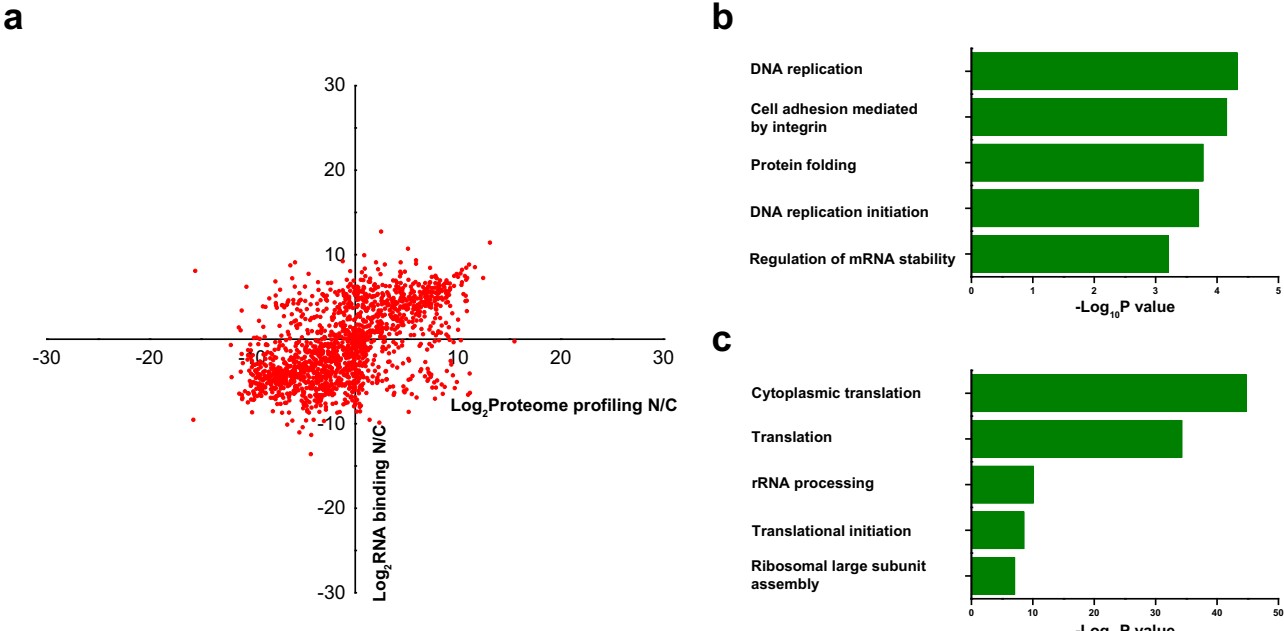

**Fig. 5 | Analysis of the subcellular distribution of RBQ correlates with the RBP abundance in nucleus and cytosol. a** Scatter plot of nucleus to cytoplasm ratio of subcellular abundance of RBPs obtained by proteome profiling VS quantity of RBPs enriched by nuclear and cytoplasm probes. Each dot represents a RBP. **b** GO biological process of the RBPs with obviously higher abundance but relatively lower RBQ in cytoplasm. **c** GO biological process of the RBPs with enhanced abundance but decreased RBQ in nucleus. Statistical analysis was performed with hypergeometric test (BH adjusted *P* values). Source data are provided as a Source Data file.

enhanced RNA binding account for the elevated nuclear RBP enrichment of ACO2 and FH. While, for CS, MDH2 and PDHA1, the main reason should be increased RNA binding. The involvement of TCA cycle in the erastin disrupted subcellular specific RNA-RBP interactions may provide new insights for investigating their possible roles in ferroptosis induced metabolism dysregulation.

Finally, we performed chromatin-associated proteins enrichment (ChEP)[68,69] before and after erastin treatment. 175 RBPs were found to also interact with chromatin and were referred to as chromatin associated-RBPs (CA-RBPs). STRING database was used to obtain highly confident interacting proteins (interaction score > 0.7) of the 175 CA-RBPs identified by both ChEP and our BLTF enrichment (Fig. 8a). The CA-RBPs were represented by nodes, and the interactions between them were indicated by edges. The protein-protein interaction (PPI) networks listed in Supplementary Data 5 contain 175 nodes and 364 edges with their interactors. Pathway and GO analysis of the total CA-RBPs showed obviously enriched DNA related functions, including (Epigenetic regulation of) gene expression, Histone deacetylate, Chromatin organization/remodeling, Nucleosome assembly and DNA replication (Fig. 8b, c). However, we also found a significant module in the PPI network (green nodes) displaying enriched RNA related functions, such as mRNA Splicing/ Spliceosome, Capped Intron-Containing Pre-mRNA, Metabolism of RNA, RNA structure unwinding and Response to exogenous dsRNA (Fig. 8d, e). Furthermore, GO terms that were not related to DNA or RNA were also find (Rhythmic process, SUMOylation and regulation of megakaryocyte differentiation), indicating complex regulating roles of the CA-RBPs. RNA binding behaviors of the 175 CA-RBPs were also investigated upon erastin induction. 25 of the CA-RBPs exhibited upregulated RNA-binding, while none of them was downregulated in the volcano plot (Fig. S20, Supporting Information). Five of the upregulated CA-RBPs, including PRDX5, SND1, KDM1A, XRCC6 and CEBPB, were reported to regulate ferroptosis[54,70–74]. For instance, SND1 promotes degradation of GPX4 by destabilizing HSPA5 mRNA and suppressing HSPA5 expression, thus promoting ferroptosis in osteoarthritis chondrocytes.

## Discussion

In this work, we described a strategy for organelle RNP enrichment based on photoinduced subcellular-specific RNA labeling. We demonstrated the ability of a series of subcellular-targeting RNA labeling probes to perform unbiased capture of RBPs bound with various kinds of RNAs in the nucleus and cytoplasm. By combination of the subcellular targeting peptide and the cycloaddition reaction between furocoumarin and uracil, the subcellular-specific RBP enrichment probes developed in this work achieved 54.4% and 85.7% increase in nuclear and cytoplasmic RBPs identification compared with previously reported methods[23,35]. Furthermore, the enrichment probes bind to RNAs at living cell level, which is advantageous to obtained more accurate subcellular location of the RBPs and their dynamic changes in biological process than via organelle purification after cells lysis, such as serIC[24].

We also observed large-scale variation in subcellular RBP enrichment using the BLTF and BETF probes upon ferroptosis induction. Interestingly, a total of 278 RBPs exhibited enhanced interactions with RNAs in the nucleus, while 602 RBPs exhibited decreased binding with RNAs in the cytoplasm during ferroptosis process. Among these regulated RBPs, translation was displayed as one of the commonly enriched GO functions by different ferroptosis inducers, indicating ferroptosis could disturb protein translation via different pathways. The RBPs exhibited both upregulation in the nuclear RNPs and downregulation in the cytoplasmic RNPs by the same ferroptosis inducer were considered as the nucleoplasmic translocation candidates. Immunofluorescence imaging and organelle-specific differential proteomics analysis revealed cytoplasm to nuclear translocation or RBQ changes of different RBPs upon ferroptosis induction, indicating capability of the probes developed in this work for quantitatively studying RNA binding variation of the RBPs in subcellular compartments upon stimulation.

Having said so, our RBP enrichment probes still have some limitations. First, the enrichment probes are not commercially available yet, so the researchers have to prepare the probes by themselves. We provided detailed synthesis protocol in the method part to facilitate

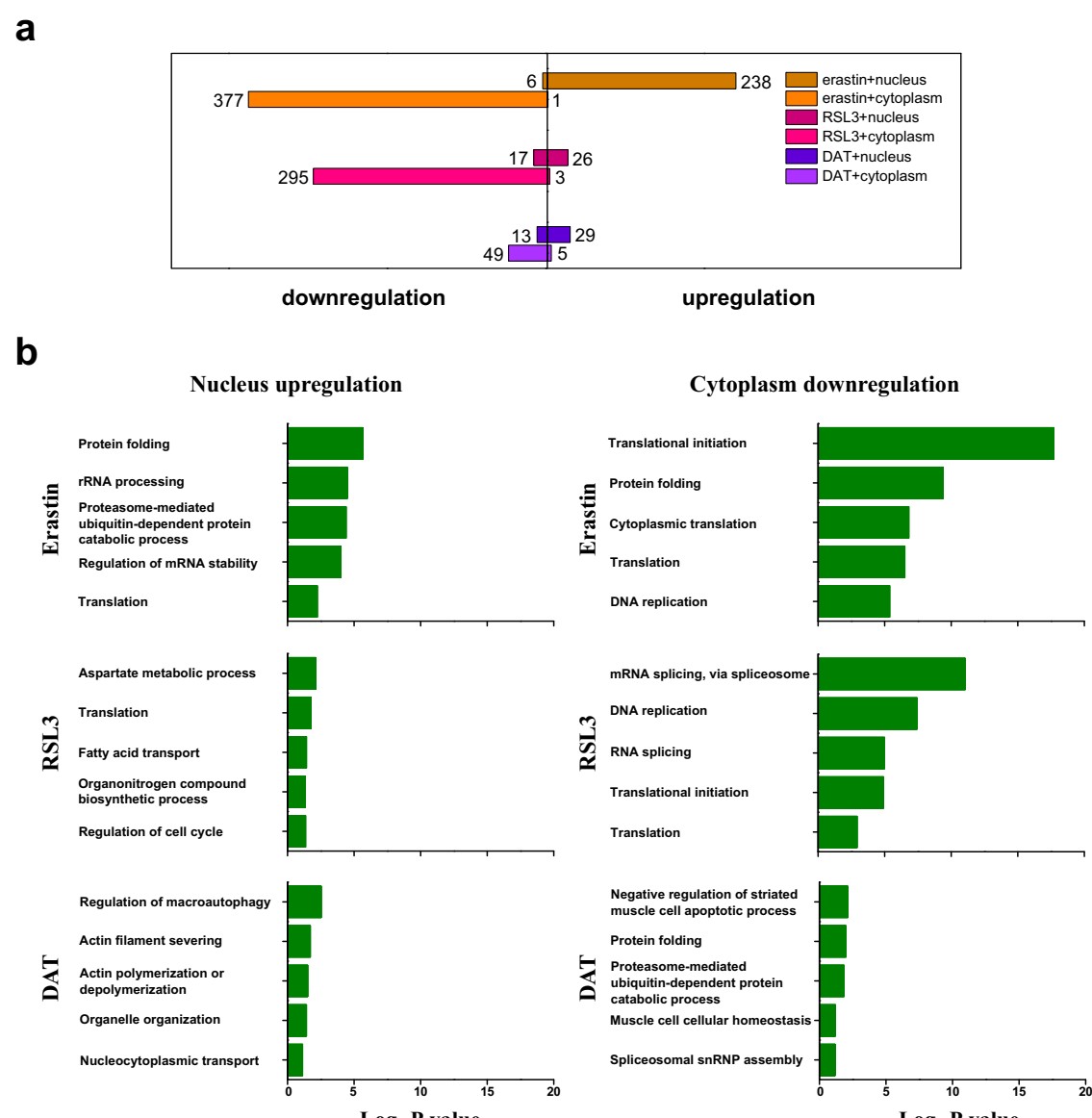

**Fig. 6 | Analysis of ferroptosis inducer regulated subcellular RBPs.**
**a** Identification map of erastin, RSL3 and DAT regulated RBPs in nuclear and cytoplasmic RNPs. **b** GO biological process of the upregulated RBPs in nuclear RNPs and downregulated RBPs in cytoplasmic RNPs by erastin, RSL3 and DAT treatment. Statistical analysis was performed with hypergeometric test (BH adjusted *P* values). Source data are provided as a Source Data file.

adoption of this enrichment method by other labs. Furthermore, currently we can only achieve nuclear and cytoplasmic level RBP enrichment. Other organelles, such as mitochondria, are also doable by this method providing the targeting moieties are available (for example triphenylphosphine for mitochondria), but organelles with higher resolution or without membrane structure may require a completely new probe designing.

## Methods

### Cell and culture conditions
HeLa cells obtained from ATCC (CCL-2) were grown in Dulbecco's modified Eagle's medium (DMEM; Gibco) supplemented with 10% FBS (Gibco), 100 μ/ml penicillin (Gibco), and 100 μg/ml streptomycin (Gibco) at 37 °C in a 5% CO2 atmosphere. Cells were tested routinely for mycoplasma contamination by PCR screening. For the ferroptosis induction experiment, HeLa cells were treated with different concentrations of erastin, RLS3 or DAT at 37 °C in DMEM (10% FBS, 100 μ/ml penicillin, and 100 g/ml streptomycin) for several hours before the cells were collected.

### Cell viability measurement
HeLa cells were seeded in a 96-well adherent plate at a density of $10^4$ per well. After treated with ferroptosis inducers with indicated time and concentration, the cells were exposed to 10 μL Cell Counting Kit-8 (CCK8, Lablead) for 1 h at 37 °C, 5% CO₂ in an incubator. The absorbance at a wavelength of 450 nm was determined using a Microplate Reader (Thermo, multiskan MK3). Cell viability under test conditions was reported as a percentage relative to the negative control. At least three wells were tested per experiment.

### Cell death measurement
HeLa cells were seeded in a 96-well adherent plate at a density of $10^4$ per well. After overnight culture, the cells were treated with 20 μM erastin for 2 h, 0.5 μM RSL3 for 4 h, or 50 μM DAT for 4 h, respectively. The cell death assay was performed using an Annexin V-FITC/ PI Apoptosis Detection Kit (MCE) and measured with a flow cytometer (BD, FACSVerse). The data were analyzed using FlowJo 10.2 software.

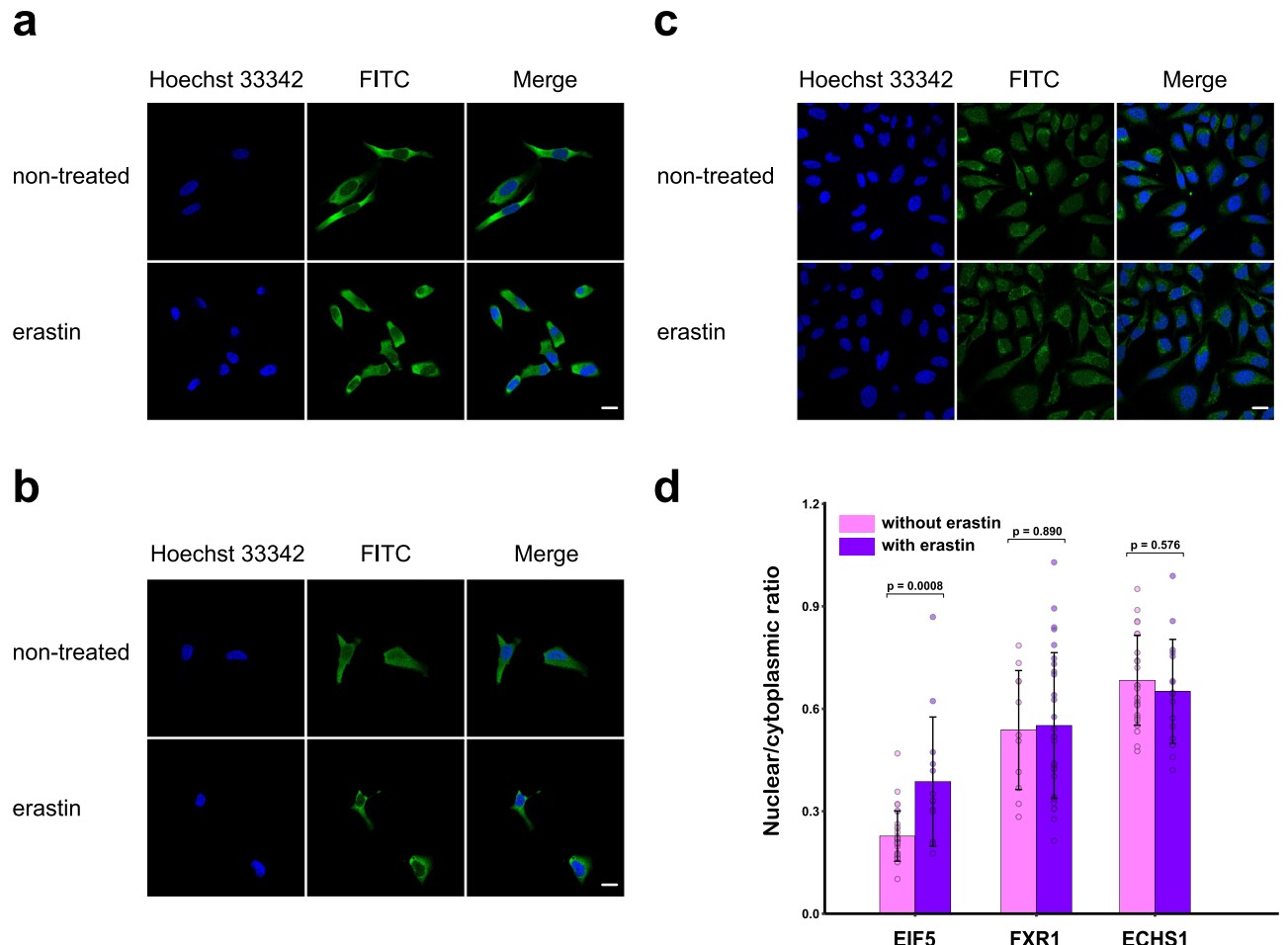

**Fig. 7 | Immunofluorescence imaging of translation-related RBPs with or without erastin induction.** Immunofluorescence images of (**a**) EIF5, (**b**) FXR1 and (**c**) ECHS1 with or without erastin induction. Scale bars, 20 µm. **d** Nuclear/ cytoplasmic fluorescence intensity ratio of EIF5, FHFXR1 and ECHS1 in (**a**–**c**). Values are the mean ± S.D. of $n = 10$ cells per condition, two-way ANOVA. Source data are provided as a Source Data file.

## Synthesis and MS characterization of the probes

The synthesis steps and characterization of the subcellular targeting probes are shown in Supplementary Fig. S1a–i, Supporting Information. Briefly, we first synthesized the furocoumarin substrate precursor N-(14-(2,5-dioxo-2,5-dihydro-1H-pyrrol-1-yl)−3,6,9,12-tetraoxatetradecyl) −4-((7-oxo-7H-furo[3,2-g]chromen-9-yl)oxy)butanamide (compound 1) by 2,5-dioxopyrrolidin-1-yl 4-((7-oxo-7H-furo[3,2-g]chromen-9-yl)oxy) butanoate (Thermo) and 1-(14-amino-3,6,9,12-tetraoxatetradecyl)−1*H*-pyrrole-2,5-dione (ToYongBio), and then reacted it with different probe substrates (SYNPEPTIDE) by cycloaddition to form various functional probes (BTF, BLF, BLTF, BL3F, BEF, BETF, BELF, BLEF). The detailed chemical equations are provided in the Fig. S1, Supporting Information.

N-(14-(2,5-dioxo-2,5-dihydro-1H-pyrrol-1-yl)−3,6,9,12-tetra-oxatetradecyl)−4-((7-oxo-7H-furo[3,2-g]chromen-9-yl)oxy)butana-mide (**1**)

1. A solution of 2,5-dioxopyrrolidin-1-yl 4-((7-oxo-7H-furo[3,2-g] chromen-9-yl)oxy) butanoate (Thermo, 19.3 mg, 0.05 mmol) in tetrahydrofuran solution (Sigma, ≥99.0%, 1 mL) was added dropwise into a 10 mL conical flask with saturated sodium bicarbonate solution (Sigma, ≥99.7%, 2 mL) of 1-(14-amino-3,6,9,12-tetraoxatetradecyl)−1H-pyrrole-2,5-dione (ToYongBio, 31.6 mg, 0.1 mmol) and stirred overnight at room temperature by a magnetic stirrer.
   Caution: Thin-layer chromatography (TLC) was applied by glass support silica gel 60 matrix (Merck) to monitor products (DCM/MeOH, 15/1, v/v as eluent).

2. The reaction mixture was transferred to a 15 mL tube. DCM (J&K Scientific, ≥99.5% 10 mL) was added to the reaction mixture to extract the products. Repeat the DCM extraction step 3 times.

3. The organic phase was combined and dried over anhydrous sodium sulfate (Sigma, ≥99.0%, 500 mg), then evaporated to dryness. The crude product was purified by silica column chromatography (DCM/MeOH, 25/1, v/v) to give the compound **1**.

Biotin-C(Furocoumarin)YGRKKRRQRRR (Biotin-TAT-Furocoumarin, BTF)

1. A solution of compound **1** (11.7 mg, 0.02 mmol) in aqueous acetonitrile solution (ACN/$H_2O$ = 3:1, v/v, 1 mL) was added into Biotin-C-TAT (SYNPEPTIDE, 18.9 mg, 0.01 mmol). The resulting solution was stirred overnight at room temperature by a magnetic stirrer. Caution: TLC was applied to monitor the products (DCM/MeOH, 10/1, v/v as eluent).

2. The solvent was removed under reduced pressure and the residue was purified by silica column chromatography (DCM/MeOH, 10/1, v/v) to give the compound BTF.

Biotin-C(Furocoumarin)PKKKRKV (Biotin-NLS-Furocoumarin, BLF)

1. A solution of compound 1 (11.7 mg, 0.02 mmol) in aqueous acetonitrile solution (ACN/$H_2O$ = 3:1, v/v, 1 mL) was added into Biotin-C-NLS SYNPEPTIDE, 12.1 mg, 0.01 mmol. The resulting solution was stirred overnight at room temperature by a magnetic stirrer.

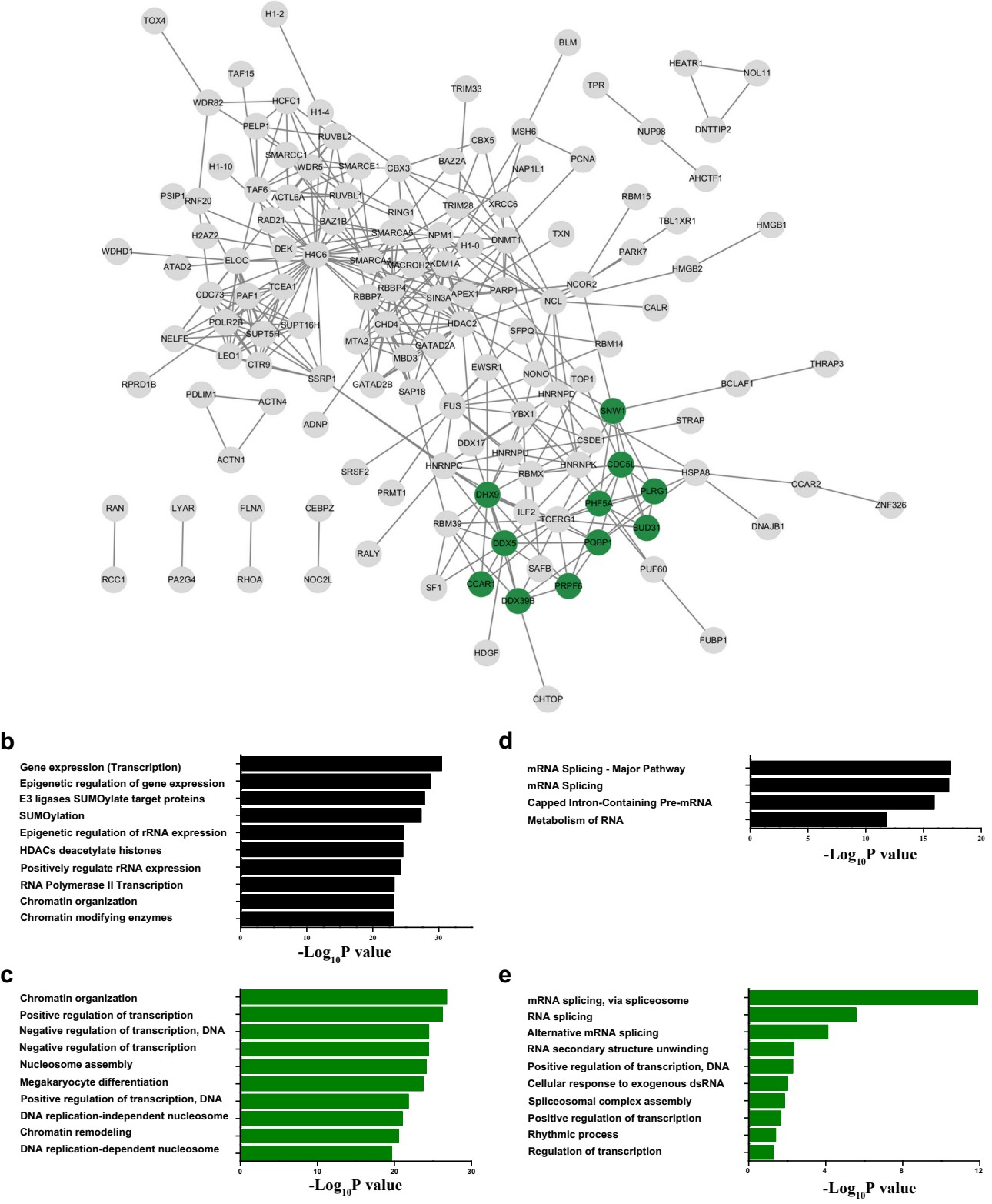

**Fig. 8 | Analysis of the 175 CA-RBPs. a** PPI network of the CA-RBPs. The green nodes represent significant module from the PPI network containing 11 nodes determined by MCODE plugin. **b** Pathway analysis of the CA-RBPs. **c** GO biological process analysis of the CA-RBPs. **d** Pathway analysis of the 11 CA-RBPs from the enriched PPI cluster by MCODE. **e** GO biological process analysis of the 11 CA-RBPs from the enriched PPI cluster by MCODE. Statistical analysis was performed with hypergeometric test (BH adjusted *P* values). Source data are provided as a Source Data file.

Caution: TLC was applied to monitor the products (DCM/MeOH, 10/1, v/v as eluent).

2. The solvent was removed under reduced pressure and the residue was purified by silica column chromatography (DCM/MeOH, 10/1, v/v) to give the compound BLF.

Biotin-C(Furocoumarin)PKKKRKVYGRKKRRQRRR (Biotin-NLS-TAT-Furocoumarin, BLTF)

1. A solution of compound **1** (11.7 mg, 0.02 mmol) in aqueous acetonitrile solution (ACN/$H_2O$ = 3:1, v/v, 1 mL) was added into Biotin-C-NLS-TAT SYNPEPTIDE, 27.5 mg, 0.01 mmol. The resulting solution was stirred overnight at room temperature by a magnetic stirrer.
Caution: TLC was applied to monitor the products (DCM/MeOH, 10/1, v/v as eluent).

2. The solvent was removed under reduced pressure and the residue was purified by silica column chromatography (DCM/MeOH, 10/1, v/v) to give the compound BLTF.

Biotin-C(Furocoumarin)PKKKRKVPKKKRKVPKKKRKV (Biotin-NLS-NLS-NLS-Furocoumarin, BL3F)

1. A solution of compound **1** (11.7 mg, 0.02 mmol) in aqueous acetonitrile solution (ACN/$H_2O$ = 3:1, v/v, 1 mL) was added into Biotin-C-NLS-TAT SYNPEPTIDE, 29.4 mg, 0.01 mmol. The resulting solution was stirred overnight at room temperature by a magnetic stirrer.
Caution: TLC was applied to monitor the products (DCM/MeOH, 10/1, v/v as eluent).

2. The solvent was removed under reduced pressure and the residue was purified by silica column chromatography (DCM/MeOH, 10/1, v/v) to give the compound BL3F.

Biotin-C(Furocoumarin)LQLPPLERLTLD (Biotin-NES-Furocoumarin, BEF)

1. A solution of compound **1** (11.7 mg, 0.02 mmol) in aqueous acetonitrile solution (ACN/$H_2O$ = 3:1, v/v, 1 mL) was added into Biotin-C-NES SYNPEPTIDE, 17.4 mg, 0.01 mmol. The resulting solution was stirred overnight at room temperature by a magnetic stirrer.
Caution: TLC was applied to monitor the products (DCM/MeOH, 10/1, v/v as eluent).

2. The solvent was removed under reduced pressure and the residue was purified by silica column chromatography (DCM/MeOH, 10/1, v/v) to give the compound BEF.

Biotin-C(Furocoumarin)LQLPPLERLTLDYGRKKRRQRRR (Biotin-NES-TAT-Furocoumarin, BETF)

1. A solution of compound **1** (11.7 mg, 0.02 mmol) in aqueous acetonitrile solution (ACN/$H_2O$ = 3:1, v/v, 1 mL) was added into Biotin-C-NES-TAT SYNPEPTIDE, 32.8 mg, 0.01 mmol. The resulting solution was stirred overnight at room temperature by a magnetic stirrer.
Caution: TLC was applied to monitor the products (DCM/MeOH, 10/1, v/v as eluent).

2. The solvent was removed under reduced pressure and the residue was purified by silica column chromatography (DCM/MeOH, 10/1, v/v) to give the compound BETF.

Biotin-C(Furocoumarin)LQLPPLERLTLDPKKKRKV (Biotin-NES-NLS-Furocoumarin, BELF)

1. A solution of compound **1** (11.7 mg, 0.02 mmol) in aqueous acetonitrile solution (ACN/$H_2O$ = 3:1, v/v, 1 mL) was added into Biotin-C-NES-NLS SYNPEPTIDE, 26.0 mg, 0.01 mmol. The resulting solution was stirred overnight at room temperature by a magnetic stirrer.

Caution: TLC was applied to monitor the products (DCM/MeOH, 10/1, v/v as eluent).

2. The solvent was removed under reduced pressure and the residue was purified by silica column chromatography (DCM/MeOH, 10/1, v/v) to give the compound BELF.

Biotin-C(Furocoumarin) PKKKRKVLQLPPLERLTLD (Biotin-NLS-NES-Furocoumarin, BLEF)

1. A solution of compound **1** (11.7 mg, 0.02 mmol) in aqueous acetonitrile solution (ACN/$H_2O$ = 3:1, v/v, 1 mL) was added into Biotin-C-NLS-NES SYNPEPTIDE, 26.0 mg, 0.01 mmol. The resulting solution was stirred overnight at room temperature by a magnetic stirrer.
Caution: TLC was applied to monitor the products (DCM/MeOH, 10/1, v/v as eluent).

2. The solvent was removed under reduced pressure and the residue was purified by silica column chromatography (DCM/MeOH, 10/1, v/v) to give the compound BLEF.

The MS spectra and molecular weight of the synthesized probes shown in Fig. S2 and Table S1 (Supporting Information) confirmed successful synthesis of the probes.

### Immunofluorescence staining of the probes

HeLa cells were plated on 35 mm glass-bottom culture dishes (Nest) and incubated overnight for adherence. The cells were then incubated with the probes for 10 min at 37 °C in a 5% $CO_2$ atmosphere, followed by irradiating with 254-nm UV light at 0.25 J/$cm^2$ on ice using a UV cross-linker (CL-1000; UVP). After washing with cold PBS (Corning) twice, the cells were irradiated with 365-nm UV light at 2 J/$cm^2$ for 4 min. For imaging, the cells were fixed in 4% formaldehyde in PBS for 30 min and incubated with FITC-SA (Thermo) for another 30 min to stain the probe-bound RNPs at room temperature, followed by washing with PBS twice. Next, the cells were incubated with Hoechst 33342 (Sigma) for 10 min to stain nuclei (Pierce, 1:1000). Fluorescence confocal microscopy imaging was performed with an A1R (×60/1.4 oil immersion objective lens; Nikon). Images were acquired by NIS-Elements AR (Nikon) software.

### Immunofluorescence staining to verify the subcellular location of RBPs

HeLa cells were plated on 35 mm glass-bottom culture dishes (Nest) and incubated overnight for adherence. For imaging, the cells were fixed in 4% formaldehyde (Beyotime) in PBS for 30 min and permeabilized for 5 min with 0.2% Triton-X 100 (Solarbio). After blocking with 5% goat serum (Solarbio) for 1 h, the cells were incubated with primary antibodies (Proteintech) at 37 °C overnight and incubated with secondary antibodies (Proteintech) for another 1 h at room temperature. Finally, the cells were incubated with Hoechst 33342 to stain nuclei for 10 min. It was necessary to wash with PBS twice between each step to remove the residue reagents from the previous step. Fluorescence confocal microscopy imaging was performed with an A1R (×60/1.4 oil immersion objective lens; Nikon). Images were acquired by NIS-Elements AR (Nikon) software. The acquired images were analyzed using ImageJ software to calculate the fluorescence intensities in the nucleus and cytoplasm.

### RNP tagging by subcellular-specific probes via living cell labeling

All buffers were prepared using RNase-free $H_2O$. HeLa cells were cultured in 150-mm dishes for 24–36 h to approximately 90% confluence. The cells were harvested with trypsin-EDTA (Gibco) and then centrifuged at $1000 \times g$ for 3 min. After washing with PBS, the cell pellets were resuspended with fresh DMEM in 12-well plates. Subcellular specific probe labeling was conducted by adding the probe to a final

concentration of 5 μM at 37 °C under 5% $CO_2$ for 10 min incubation. After washing twice with cold PBS, the cells were irradiated with 254-nm UV light at 0.25 J/cm$^2$ on ice using a UV cross-linker (CL-1000; UVP). After washing twice with cold PBS, the cells were irradiated with 365-nm UV light for 4 min on ice. The cells were washed twice with cold PBS and resuspended in 250 μL lysis buffer containing 0.5% SDS with protease inhibitor (Roche) and ribonucleoside vanadyl complex (New England Biolabs). The suspension was homogenized by a 26-gauge needle and diluted with 1 mL of PBS that contained protease inhibitor and ribonucleoside vanadyl complex. The cell lysate was centrifuged at $16,000 \times g$ for 15 min, and the supernatant was transferred to a new RNase-free tube.

## Isolation of RNPs by streptavidin beads

The cell lysate was mixed with 100 μl precleared streptavidin magnetic beads (Thermo) and then incubated with gentle rotation at 4 °C for 1 h and then incubated with 20 μL PBS at 37 °C for 1 h after discarding the supernatant. The beads were subsequently washed twice with 200 μL of 0.2% SDS, twice with 200 μL of 8 M urea and twice with 200 μL of 50 mM $NH_4HCO_3$. Then, the beads were incubated with 20 μL of 0.01 mg/ml RNase A at 37 °C for 1 h to elute RBPs. The eluted RBPs were clarified by centrifuging at $12,000 \times g$ for 10 min, and the supernatant was collected for SDS−PAGE analysis with silver staining or MS-based proteomic analysis. For the control samples in MS-based proteomic analysis, the cells and lysates were subjected to the same treatment except that the beads were incubated with 20 μL of 0.1 mg/ml RNase A at 37 °C for 1 h before the washing step, and the washing solution was discarded.

## RNA sequencing of the isolated RNA

The isolation of RNPs from HeLa cells was carried out through streptavidin magnetic bead enrichment, utilizing the method described above. The bead-bound RNPs were incubated with 400 μL elution buffer (12.5 mM biotin, 75 mM NaCl, 7.5 mM Tris−HCl, 1.5 mM EDTA, 0.15% SDS, 0.075% sarkosyl and 0.02% Na-deoxycholate dissolved in RNase-free $H_2O$) at RT for 20 min, followed by an additional 10 min at 65 °C. The supernatant was collected, and the beads were eluted once more to obtain an 800 μL solution in total. An equal volume of proteinase buffer (100 mM Tris−HCl, 12.5 mM EDTA, 150 mM NaCl and 2% SDS dissolved in RNase-free $H_2O$) with 2 mg/mL Proteinase K (Ambion) was mixed with the supernatant and incubated at 55 °C for 1 h. The eluted RNAs were further purified using TRIzol (Ambion) in accordance with the manufacturer's instructions.

For the control experiments extracting cytoplasmic and nuclear RNA using density gradient centrifugation, the cells were harvested with trypsin-EDTA and then centrifuged at $1000 \times g$ for 3 min. After washing with PBS, the cells were resuspended in NP-40 lysis buffer (10 mM Tris pH 7.4; 150 mM NaCl; 0.15% NP-40; 1 mM DTT dissolved in RNase-free $H_2O$). After 5 min of incubation on ice, the cells were gently placed on a cold sucrose cushion and centrifuged for 10 min at 1000 g. The supernatant was collected as the cytoplasm, and the precipitate was the nucleus. The separated cytoplasm RNAs and nuclear RNAs were further purified by TRIzol following the manufacturer's instructions.

## SDS-PAGE analysis of the isolated RBPs

The supernatants containing the released RBPs were transferred to a new tube and diluted with 60 μL of 50 mM $NH_4HCO_3$. The solution was denatured with 20 μL of 5X loading buffer supplemented with 6 μL of β-mercaptoethanol at 95 °C for 10 min. Five percent of the whole-cell lysate was used as the input control samples. The resulting samples were resolved on a 10% SDS-PAGE gel and were then visualized by a Protein Silver Stain Kit (Cwbio).

## Proteome profiling of HeLa cells

HeLa cells were cultured in 150-mm dishes for 24−36 h to approximately 90% confluence. The cells were harvested with trypsin-EDTA

(Gibco) and then centrifuged at $1000 \times g$ for 3 min. After washing with PBS, the cell pellets were resuspended in lysis buffer containing 0.5% SDS with protease inhibitor (Roche). The suspension was homogenized by a 26-gauge needle and diluted with 1 mL PBS containing protease inhibitor. The cell lysate was centrifuged at $16,000 \times g$ for 15 min, and the supernatant was transferred to a spin filter column.

## Sample preparation for MS-based proteomic analysis

The samples for MS analysis were obtained by the filter-aided sample preparation method (FASP)[75]. Briefly, the supernatants containing the released RBPs were transferred into a spin filter column (10 kDa). The samples were reduced by 10 mM TCEP at RT for 30 min and alkylated by 50 mM CAA at RT for 30 min in the dark. Subsequently, the buffer was exchanged with 50 mM $NH_4HCO_3$ three times, and the RBPs were digested by trypsin at a ratio of 50:1 at 37 °C for 16 h. The obtained peptides were collected via filtrate by centrifuging at $14,000 \times g$ for 20 min. After vacuum drying, the peptides were then dissolved in 200 μL of 100 mM triethyl ammonium bicarbonate (TEAB, Sigma). Stable isotopic dimethyl labeling was conducted for quantitative comparison between the experimental groups of the RBPs and the control groups[76]. Briefly, the experimental groups (200 μL each) were treated with 8 μL of 4% (vol/vol) $CD_2O$ (Sigma), while the control groups (200 μL each) were treated with 8 μL of 4% (vol/vol) $CH_2O$ (Sigma). The solutions of experimental and control groups were both incubated with 8 μL 0.6 M $NaBH_3CN$ (Sigma) on a shaker at room temperature for 1 h, followed by the addition of 32 μL of 1% (vol/vol) ammonia solution. After adding 16 μL of formic acid, the corresponding medium ($CD_2O + NaBH_3CN$) and light ($CH_2O + NaBH_3CN$) isotopically labeled experimental and control samples were mixed and then subjected to StageTip C18 desalting before MS analysis.

## Proteomic identification by LC−MS/MS

The FASP-digested and dimethyl-labeled peptides were vacuum dried and redissolved in 0.1% FA. The samples were separated by a homemade 15 cm length reversed-phase column (150 μm id) packed with Ultimate XB-C18 1.9 μm resin (Welch materials). An Easy nLC 1200 system (Thermo) was used to fractionalize the peptides at a flow rate of 600 nL/min according to the following gradient: 7−12% B for 6 min, 12−30% B for 51 min, 30−45% B for 10 min, 45−95% B for 1 min, and 95% B for 7 min (solvent A was 0.1% formic acid, solvent B was 0.1% formic acid in 80% acetonitrile). The LC was coupled to an Orbitrap Fusion™ Tribrid™ mass spectrometer (Thermo) via a nanoelectrospray ionization source. Full-scan mass spectra were acquired in the Orbitrap (scan range 300−1400 m/z, 120,000 resolution, maximum injection time 100 ms and AGC target value of 5e5) in data-dependent acquisition mode, followed by Higher-energy Collision Dissociation (HCD) with 32% normalized collision energy. The ion trap was used to acquire MS2 detection with the top 20 MS/MS scans using higher-energy collision dissociation (HCD) at 32% normalized collision energy. The AGC target was set to 5e3, and the maximum injection time was 35 ms. The target ions selected for MS/MS were dynamically exclusion within 18 s.

## Analysis of RNA sequencing data

For RNA library preparation, the TruSeq RNA Library Prep Kit v2 (Illumina, not stranded) was used with the extracted RNA samples. The samples were barcoded to be sequenced in one lane on a HiSeq2500. Ribosomal RNA (rRNA) was removed from the extracted total RNA using a SEQuoia RiboDepletion kit (Bio-Rad). To estimate the content of RNA biotypes in libraries, reads were aligned to the hg38 assembly using STAR. Subsequently, HTSeq-count was used to perform counting process, which relied on the reference genomes of human (release 34, GRCh38) annotated by GENCODE and the GTF feature 'gene' for counting.

## Analysis of proteomic MS data

The MS raw files were searched using MaxQuant (version 2.4.2.0) against the UniProt database (release on 2020, 20375 entry). The search parameter digestion enzyme was set as trypsin allowing a maximum of two missed tryptic cleavages, with the minimal peptide length as six amino acids. Carbamidomethyl cysteine was selected as a constant modification, while methionine oxidation and acetyl N-terminal were allowed as variable modifications. For peptide identification, the mass tolerances for precursor ions and fragment ions were set to 20 ppm and 0.5 Da, respectively. A threshold of ≤1% was allowed for both the peptide false discovery rate (FDR) and protein FDR. Imputation of missing values was performed by deterministic minimum imputation strategy in each dataset[77,78]. For RBP filtering, "Perseus" software and Student's T test were used, and 1% FDR was applied. Proteins that considered as RBPs were identified based on the following criteria: a minimum of two unique peptides in at least two parallel tests, a fold change of two or greater, and $P < 0.01$ in the experimental groups compared with the control groups. GO and pathway analyses were conducted using Funrich Version 3.1.3 and Reactome[79,80]. Protein domain analysis was performed using SMART (http://smart.embl-heidelberg.de/) against the PFAM and SMART domain databases[81]. Global protein sequence features were computed using the R (Version 4.0.3) package 'peptides' with the scales 'Kyte-Doolittle' for hydrophobicity and 'EMBOSS' for isoelectric point. The intrinsic disorder of proteins was derived using IUPred (https://iupred2a.elte.hu/)[82]. Disordered amino acid residues were defined by an IUPred score of 0.4, and the fraction of disordered amino acid residues was computed for each protein[83]. The percentage of positively charged amino acids was calculated by Python (Version 3.8).

## Reporting summary

Further information on research design is available in the Nature Portfolio Reporting Summary linked to this article.

## Data availability

The mass spectrometry proteomics data have been deposited to the ProteomeXchange Consortium [http://proteomecentral.proteomexchange.org] via the iProX partner repository with the dataset identifier PXD033927 [https://www.iprox.cn/page/project.html?id=IPX0004457000]. All RNA-seq data used in this manuscript have been deposited in Gene Expression Omnibus [www.ncbi.nlm.nih.gov/geo]. The data about the type and relative distribution of the isolated RNAs were deposited under accession number GSE205553. All other data supporting the findings of this study are available from the corresponding author upon request. Source data are provided as a Source Data file with this paper. Source data are provided with this paper.

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

## Acknowledgements

This study was supported by the National Key R&D Program of China (No. 2021YFA1302604 to W.Q.), the National Natural Science Foundation of China (No. 32088101, 32371504 to W.Q.), the National Key Laboratory of Proteomics (No. SKLP-K201706 to W.Q.). The authors thank Proteomics Technological Platform, Imaging Facility (Ms. Chunhua Zhang and Mr. Jun Chen) and Mr. Youdong Xu of National Center for Protein Sciences Beijing (NCPSB) for their assistance.

## Author contributions

Weijie Qin conceived the project. Haofan Sun designed and performed most experiments. Bin Fu was responsible for the output of mass spectrometry data. Weijie Qin, Ping Xu and Xiaohong Qian supervised all experiments. Weijie Qin and Haofan Sun discussed results and wrote the manuscript.

## Competing interests

The authors declare no competing interests.
