## [Peer Review File · Nature Communications]

REVIEWER COMMENTS

Reviewer #1 (Remarks to the Author):

Qin and coworkers developed a chemoproteomic method to identify subcellular RNA-binding proteins (RBPs). The essential element is a trifunctional probe with three parts: 1) a furocoumarin group to react with uracil upon 365nm UV; 2) a subcellular localization sequence to guide the probe to a targeted subcellular organelle; 3) a biotin handle for enrichment. By applying 254nm UV irradiation to crosslink nucleic acids with proteins first and then using the probe to capture RNAs in the specific subcellular region, they were able to enrich RBPs in the targeted organelle, in this case, nucleus and cytoplasm, for proteomic analysis. The experiments resulted in discovery of a large number of subcellular RBPs. The authors further applied the method in analyzing RBPs dynamics between these two regions upon induction of cell ferroptosis. Overall, the technical part of this work is novel and the data quality of the subcellular RBP profiling looks good. However, the ferroptosis part seems quite detached and not much biological insights are obtained except a couple of new datasets. Based on the rebuttal letter provided in the submission, it seemed that the authors had done some phosphorylation profiling for the application of their subcellular RBP profiling method and upon the editor's initial comment, they changed it to ferroptosis? What is the logic behind that? Any literature supporting the relevance between RBPs and ferroptosis? How does the current data contribute to our understanding of how ferroptosis is regulated? In my option, the authors should focus on verifying some novel RBPs revealed by their method given the subcellular resolution, instead of making vague links to a "hot-topic" cell death type. In addition, the figure quality needs much improvement. In particular, the reviewer is not convinced that a panel of scatter plots is very informative in figure 2. Minor: page 8, um is not a unit for concentration but rather for length (nanometer).

Reviewer #2 (Remarks to the Author):

In this paper, Sun et al. present a comprehensive chemoproteomic profiling of nuclear and cytoplasmic RNA-binding proteins (RBPs) using organelle-targeting furocoumarin-biotin probes. This method represents a significant technical advancement compared to previous approaches for mapping nuclear RBPs. Furthermore, the authors successfully apply this method to investigate the subcellular-specific RBPs involved in ferroptosis, providing valuable insights into the regulation of RNA-protein interactions during this cellular process. The study also uncovers potential candidates for nucleoplasmic translocation, which are further confirmed through immunofluorescence imaging.

Overall, this novel method captures broad interest within the scientific community. The datasets containing subcellular RBPs, particularly those related to ferroptosis, are of immense value and

contribute to our understanding of protein trafficking during this cellular phenomenon. Therefore, I highly recommend this publication for consideration in Nature Communications, with only minor revisions.

1. The label "RNase A washing" in Fig. 1d is misleading since RNase A washing is used as a control in Fig. 1e. It would be more appropriate to label it as "RNase A elution" in Fig.1d.
2. As mentioned in the original paper, the authors should change "APEX-OAPS" to "APEX-PS" to maintain consistency.
3. In order to better understand the advantages of this subcellular-specific RNA labeling method, I suggest that the authors further compare it with previous datasets obtained from serIC, RBR-ID, and APEX-PS. This would provide a more comprehensive evaluation of the method's strengths in subcellular-specific RBP profiling.

Reviewer #3 (Remarks to the Author):

The manuscript by Haofan Sun and colleagues presents a novel method utilizing furocoumarin-containing probes to selectively label and isolate nuclear or cytoplasmic RNAs. Through in-cellulo UV-crosslinking of proteins and RNA, the researchers isolated ribonucleoprotein complexes from the nucleus and cytoplasm. The study identified a significant number of nuclear RBPs (1221) and cytoplasmic RBPs (1333), allegedly surpassing previous techniques. Moreover, the authors investigated changes in the nuclear and cytosolic RBPomes following treatment with three ferroptosis inducers. They identified numerous responsive proteins including 59 RBPs defined as "nucleoplasmic translocation candidates". Overall, this approach offers an intriguing opportunity for globally characterizing RBPs with subcellular resolution across biological contexts.

However, there are a few areas that require attention. Firstly, the lack of commercially available probes could limit the wider applicability of the method. Secondly, the authors should include instrumental controls to assess the technique's specificity. Lastly, it would greatly enhance the study's value to explore the biological significance of observed responses or investigate certain aspects of the underlying mechanisms. Addressing these points would further strengthen the overall findings and implications of the research.

Major comments:

- To distinguish genuine RNA-binding proteins from proteins that may associate non-specifically (e.g. independent of UV) or indirectly (e.g. through interaction with RBPs) with RNA, it is crucial to include a non-UV control in the experimental design. This control would involve treating the samples identically but without UV crosslinking at 254 nm (the irradiation at 365 nm would still be performed). By comparing the results obtained from the UV-crosslinked samples to those of the non-UV controls (generated in parallel), the specificity of the technique could be assessed in a way that the current control (RNase treatment before the washing step) does not allow. This control should be performed at least with the most effective nuclear and cytosolic probes.

- The described protocol is likely be of interest to laboratories with expertise in RNA/RBP biology but not necessarily with a strong background in chemistry. It is advisable for the authors to provide a detailed step-by-step protocol that includes comprehensive information to facilitate the procedure for non-experts. This should include specific instructions on the required equipment, reaction conditions, recommended reagent brands and their purity, as well as any additional comments and quality controls necessary for successful probe synthesis.

- The reported translocation of TCA enzymes to the nucleus following treatment with ferroptosis inducers is indeed intriguing. Validating and further investigating this observation would certainly enhance the impact of the study. While conducting an exhaustive exploration of the biological significance may be beyond the scope of the work, there are numerous relevant questions that could be addressed within a reasonable time frame. Please find below some suggestions on that line, that I have grouped into three categories:

i) Characterizing the behaviour of the probes and the overall effects of the employed treatments with ferroptosis inducers in cellular biology: Can the BETF probe penetrate mitochondria? Do the described treatments affect the integrity of mitochondria or the nucleus? It is important to confirm that the treatments do not alter the subcellular localization of the BLTF/BETF probes (Figure 6a and Figure S8).

ii) Validating the translocation of TCA proteins to the nucleus through orthogonal assays: This can be assessed through imaging, as shown in Figure 7a-c, and by performing subcellular fractionation followed by western blotting/proteomics.

iii) Investigating mechanistic aspects: Do TCA enzymes bind RNA inside or outside mitochondria? Which pool, inside or outside mitochondria, translocates to the nucleus? Is the translocation linked to changes in posttranslational modifications of the proteins? Inside the nucleus, do the proteins associate with chromatin? What are their protein interactors, and how do they change upon treatment? I understand that identifying the RNA interactors through methods like RIP or CLIP could potentially offer valuable insights. However, I acknowledge that these techniques may present challenges in obtaining clean and meaningful data. Therefore, while it could be advantageous to explore this aspect, I do not consider it a necessary requirement for the study.

- Another intriguing aspect that could be explored in a similar manner is the potential enhanced interaction of translation-related RBPs with RNAs in the nucleus following treatment with ferroptosis inducers. I suggest to validate this response and the alleged modulation of the nucleoplasmic translocation of these proteins.

- The overall writing style of the paper could be improved.

Minor comments:

- The expression “organelle-specific” in the title gives the false impression that the work covers multiple organelles. I propose to use instead something along the line of: “nuclear- and cytoplasmic-specific RNA Tagging...”

- “First, the RNA tagging probes were introduced during cell culture...” I find this expression a bit confusing.

- “Figure 1f and 1g shows that overlapping but distinct RBPs were identified by different nucleus/cytoplasm-targeting probes.” Considering that the probes are fairly similar, why the overlap of captured RBPs (among nuclear probes on one side and among cytosolic probes on the other) is relatively modest? Can the authors discuss that? Moreover, can the authors address limitations and strengths of the described probes/approach in the discussion?

- How were the “experiment” samples (Fig 1e, not treated with RNase prior wash) maintained during the RNase treatment of the control samples? One important aspect to consider is to incubate the “experiment” samples under the same conditions as the control samples, including using the same buffer and maintaining the same temperature (e.g., 37 C for 1 hour). This is essential to ensure that any observed differences in protein elution are due to the digestion of RNA and not influenced by other factors such as temperature-induced effects.

- Regarding the analysis of the nuclear and cytoplasmic RNAs (Figure 4):

- i) Quantify the number of reads mapping to intronic and exonic regions in the nuclear and cytoplasmic RNA pools. Intronic regions should be enriched in the nuclear RNA.
- ii) It might be useful to make use of the RNAseq data to quantify the reads mapping to genes and intergenic regions, in order to assess the level of contamination with genomic DNA in the nuclear and cytosolic pools.
- iii) It would be useful to know which fraction of the RNA isolated from nucleus and cytoplasm corresponds to rRNA. This is not observable in the RNAseq data as ribodepletion was applied, but could be easily estimated by capillary electrophoresis (i.e. bioanalyzer).
- iv) In Figures 4 e and f, the RNAs below 200 nt could not only correspond to tRNAs, but also to other small RNAs such as 5 S rRNA, 5.8 S rRNA, snoRNAs and snRNAs.

- It would be useful to include a second volcano plot similar to the one shown in Figure 5f but corrected by the background. In other words, the suggested volcano plot would represent the ratio in BLTF vs BETF of the ratios shown in Figure 2 FC. This means:

$\log_2 \text{fold-change}(\text{FC-BLTF vs FC-BETF}),$

with FC-BLTF and FC-BETF defined as: $\log_2 \text{fold-change}(\text{experiment group vs control group}).$

For the reasons explained above, the control groups should be non-irradiated samples, rather than the samples treated with RNase prior the wash.

- When determining the subcellular location for RNA binding of the identified RBPs (Figure 5f), if possible, it would be useful to determine the overall abundance of the proteins in nucleus or cytosol, in order to define how RNA binding activity correlates with protein abundance in each compartment. Likewise, the total proteome analysis following treatment with ferroptosis inducers (Figure S7) would greatly benefit if they would be done independently in nucleus and cytosol.

- Figure 7: Show a negative control (a protein whose nuclear/cytoplasmic ratio does not respond to the treatment with the ferroptosis inducers). Furthermore, the cytoplasmic area seems to shrink in the treated cells, is this the case? If so, how does this affect the analysis and quantification performed?

- Figure S10: Concerning the 59 “nucleoplasmic translocation candidates”, I suggest to add a table with their names and the fold change in nucleus and cytoplasm upon erastin treatment. Likewise, I would add the names of the identified responsive proteins in each of the GO clusters shown.

REVIEWER COMMENTS

Reviewer #1 (Remarks to the Author):

Qin and coworkers developed a chemoproteomic method to identify subcellular RNA-binding proteins (RBPs). The essential element is a trifunctional probe with three parts: 1) a furocoumarin group to react with uracil upon 365nm UV; 2) a subcellular localization sequence to guide the probe to a targeted subcellular organelle; 3) a biotin handle for enrichment. By applying 254nm UV irradiation to crosslink nucleic acids with proteins first and then using the probe to capture RNAs in the specific subcellular region, they were able to enrich RBPs in the targeted organelle, in this case, nucleus and cytoplasm, for proteomic analysis. The experiments resulted in discovery of a large number of subcellular RBPs. The authors further applied the method in analyzing RBPs dynamics between these two regions upon induction of cell ferroptosis. Overall, the technical part of this work is novel and the data quality of the subcellular RBP profiling looks good. However, the ferroptosis part seems quite detached and not much biological insights are obtained except a couple of new datasets. Based on the rebuttal letter provided in the submission, it seemed that the authors had done some phosphorylation profiling for the application of their subcellular RBP profiling method and upon the editor's initial comment, they changed it to ferroptosis? What is the logic behind that? Any literature supporting the relevance between RBPs and ferroptosis? How does the current data contribute to our understanding of how ferroptosis is regulated? In my option, the authors should focus on verifying some novel RBPs revealed by their method given the subcellular resolution, instead of making vague links to a "hot-topic" cell death type. In addition, the figure quality needs much improvement. In particular, the reviewer is not convinced that a panel of scatter plots is very informative in figure 2. Minor: page 8, μm is not a unit for concentration but rather for length (nanometer).

Thanks for the reviewer's positive comment on the novelty of our method and data quality of the subcellular RBP profiling. We appreciated the reviewer's recognition of the potential of our method and the valuable suggestion on further exploring its application in discovering biological insights of ferroptosis.

Programmed cell death (PCD) plays a vital role in homeostasis maintenance and diseases development^{1,2}. Literatures showed extensive regulatory relationship between RNA-binding proteins (RBPs) and PCD. For ferroptosis, different RBPs, such as GPX4, RBMS1 or ZFP36/TTP that promote or protect cells against ferroptosis were reported, indicating the complex role of RBPs as essential regulators in ferroptosis³⁻⁵. Furthermore, ferroptosis was reported to be associated with nucleocytoplasmic transport of proteins, which is intimately tied to RBP functions, as post-transcriptional regulations are often carried out in subcellular compartments⁶⁻⁹. For other types of PCD, autophagy was found to mediate transport, secretion, and decay of various kinds of RBPs¹⁰. On the other hand, splicing factors like SRSF1 regulates apoptosis and proliferation to promote mammary epithelial cell transformation¹¹. It was also reported

that RBP METTL14 suppressed pyroptosis and diabetic cardiomyopathy by downregulating TINCR lncRNA¹². Furthermore, RBP N4BP1, SMG7, PTBP1 and PCBP1 are all necroptosis regulators¹³.

However, previous works were all focused on individual RBPs and were lack of omics-level investigation on alternation of RBP-RNA interaction and nucleocytoplasmic distribution during PCD process was conducted. Therefore, we intended to systematically investigate the interaction between RNA and RBPs and the changes of their subcellular location in PCD using the proposed organelle specific RBPs enrichment probes. In the initial attempt, we applied our RBPs enrichment probes to study the changes of RBPs in staurosporine induced phosphorylation inhibition and autophagy. We discovered dozens of RBPs changed their interaction with RNAs in nucleus and cytoplasm in this process. However, since staurosporine is both an autophagy inducer and a multi-kinase inhibitor, the autophagy process is accompanied by large-scale inhibition of protein phosphorylation. As a result, it was difficult to determine the decisive factor leading to the altered RNA-RBP interaction, autophagy or phosphorylation. To better investigate the interplay of RNA and RBP in PCD, we further studied the subcellular level changes of RBPs in ferroptosis. Interestingly, large-scale RBPs displayed enhanced interaction with RNAs in nucleus but reduced association with RNAs in cytoplasm in ferroptosis induced by erastin treatment with translation related terms obviously enriched in GO biological process (Figure R1 below and Figure 6 in the manuscript). Furthermore, we discovered dozens of nucleoplasmic translocation candidate RBPs upon ferroptosis induction. The enrichment of Tricarboxylic acid (TCA) cycle and translation in the translocation candidate RBPs may provide new insights for investigating their possible roles in ferroptosis induced metabolism and protein expression dysregulation (Figure R2 below and Figure S15 in the supporting information). The above discovery was further studied using quantitative proteomic and immunofluorescence imaging analysis shown below.

a

b

Figure R1. Analysis of ferroptosis inducer regulated subcellular RBPs.

a Identification map of erastin, RSL3 and DAT regulated RBPs in nuclear and cytoplasmic RNPs.

b GO biological process of the upregulated RBPs in nuclear RNPs and downregulated RBPs in cytoplasmic RNPs by erastin.

Figure R2. Analysis of the erastin-induced nucleoplasmic translocation candidate RBPs.

a Pathway analysis of the candidates.

b GO biological process analysis of the candidates.

Following the reviewer's suggestion, we further explored and validated the above discovery in TCA and translation related variation of subcellular RBP distribution and RNA binding. As shown in the volcano plot in Figure R3, majority of the nucleus proteins (>90%) did not show obvious abundance change upon inducing ferroptosis. The five TCA proteins that were upregulated in nuclear RBP enrichment after erastin treatment were chosen for further analysis. ACO2 and FH were upregulated, while CS and MDH2 were unchanged after inducing ferroptosis in Figure R3. The remaining one (PDHA1) was not identified in the proteome of nuclear fraction. Consistent results were obtained using immunofluorescence imaging in Figure R4. ACO2 and FH were indeed translocated to the nucleus with 31.1% and 19.2% fluorescence enhancement after erastin treatment, while changes in nucleocytoplasmic distribution of CS, MDH2 and PDHA1 were negligible. We further normalized the quantitative changes of nuclear RBP enrichment by their corresponding nuclear protein abundance variation. All the four TCA related RBPs still displayed upregulated RNA binding after normalization

(Figure R5). The above results revealed that both increased protein abundance by translocation to nucleus and enhanced RNA binding account for the elevated nuclear RBP enrichment of ACO2 and FH. While, for CS, MDH2 and PDHA1, the main reason should be increased RNA binding. The above results were added in the revised manuscript, page 20 and supporting information (Figure S17-19).

Figure R3. Quantitative analysis of nuclear proteins in cells treated with or without erastin.

Volcano plot displaying log₂ fold change (FC) (x-axis) and $-\log_{10}$ P values (y-axis) of the nuclear proteome of the erastin-treated and nontreated cells. Proteins with significant increase after erastin treatment were depicted in red, and those significantly decreased after erastin treatment were depicted in blue.

Figure R4. Immunofluorescence imaging of TCA proteins with and without erastin induction.

(a)-(e) Immunofluorescence images of (a) ACO2, (b) FH, (c) CS, (d) MDH2 and (e) PDHA1 with and without erastin induction. (f) The nuclear/cytoplasmic fluorescence intensity ratio of ACO2, FH, CS, MDH2 and PDHA1 with and without erastin induction. Values are the mean \pm S.D. of at least 10 cells per condition.

Figure R5. Quantitative analysis of the BLTF probe enriched nuclear RBPs in cells treated with or without erastin after normalization by the corresponding protein abundance variation.

Volcano plot displaying \log_2 fold change (FC) (x-axis) and $-\log_{10}$ P values (y-axis) of the nuclear RBPs enriched from the erastin-treated and nontreated cells after normalization by the nuclear proteome. Significantly increased nuclear RBPs after erastin treatment were depicted in red, and significantly decreased ones were depicted in blue.

We then performed chromatin-associated proteins enrichment (ChEP)^{14,15} before and after erastin treatment. 175 RBPs were found to also interacted with chromatin and were referred to as chromatin associated-RBPs (CA-RBPs). STRING database was used to obtain highly confident interacting proteins (interaction score > 0.7) of the 175 CA-RBPs identified by both ChEP and our BLTF enrichment in Figure R6. The CA-RBPs were represented by nodes, and the interactions between them were indicated by edges. The protein-protein interaction (PPI) network containing 175 nodes and 364 edges with their interactors were listed in SupplementaryData6. Pathway and GO analysis of the total CA-RBPs showed obviously enriched DNA related functions, including (Epigenetic regulation of) gene expression, Histone deacetylate, Chromatin organization/remodeling, Nucleosome assembly and DNA replication (Figure R7). However, we also found a significant module in the PPI network (green nodes) displaying enriched RNA related functions, such as mRNA Splicing/Spliceosome, Processing of Capped Intron-Containing Pre-mRNA, Metabolism of RNA, RNA structure unwinding and Response to exogenous dsRNA (Figure R8). Furthermore, GO terms that not related to DNA or RNA were also find (Rhythmic process, SUMOylation and regulation of megakaryocyte differentiation) indicating complex regulating roles of the CA-RBPs.

RNA binding behaviors of the 175 CA-RBPs were investigated upon erastin induction. As shown in the volcano plot in Figure R9, 25 of the CA-RBPs exhibited upregulated RNA-binding, while none of them downregulated. 5 of the RNA-binding upregulated CA-RBPs, including PRDX5, SND1, KDM1A, XRCC6 and CEBPB, were reported to regulate ferroptosis^{4,16-20}. For instance, SND1 promotes the degradation of GPX4 by destabilizing the HSPA5 mRNA and suppressing HSPA5 expression, thus promoting ferroptosis in osteoarthritis chondrocytes. The above results were added in the revised manuscript, page 22.

Figure R6. PPI network of the CA-RBPs. The green nodes represent significant module from the PPI network containing 11 nodes determined by MCODE plugin.

Figure R7. Analysis of the 175 CA-RBPs. a Pathway analysis of the proteins. b GO biological process analysis of the proteins.

Figure R8. Analysis of the 11 CA-RBPs from the enriched PPI cluster by MCODE. a Pathway analysis of the proteins. b GO biological process analysis of the proteins.

Figure R9. Quantitative analysis of the CA-RBPs in cells treated with or without erastin. Volcano plot displaying \log_2 fold change (FC) (x-axis) and $-\log_{10}$ P values (y-axis) of the CA-RBPs quantified from the erastin-treated and nontreated cells. CA-RBPs with significant increase after erastin treatment were depicted in red, and significantly decreased after erastin treatment were depicted in blue.

For the 24 translation-related RBPs exhibited enhanced interaction with RNAs in nucleus after erastin induced ferroptosis, only EIF5 displayed upregulation in the nuclear proteome profiling dataset. To further validate the nucleoplasmic translocation of the translation-related RBPs, we chose one RBP as the representative for each of the three situations in our study. EIF5 represented upregulation in both RBP enrichment by the nuclear BLTF probe and proteome profiling in the nucleus fraction after erastin treatment. FXR1 represented upregulation only in RBP enrichment, while the protein abundance remained unchanged in proteome profiling. Besides the above two situations, we also chose ECHS1 as a negative control to represent RBPs that were unchanged in both RBP enrichment and proteome profiling. As shown in Figure R10, immunofluorescence imaging displayed consistent results with the proteome profiling data of the nuclear fraction. The results demonstrated that EIF5 translocated to nucleus with 72.7% fluorescence enhancement after erastin induction. FXR1 and ECHS1 showed no abundance change in the nucleus in Figure R10d. Therefore, the increased FXR1 in RBP enrichment after erastin treatment should be attributed to the enhanced binding with the corresponding RNAs in the nucleus. The above results were added in the revised manuscript, page 20.

Figure R10. Immunofluorescence imaging of translation-related RBPs with or without erastin induction. (a)-(c) Immunofluorescence images of (a) EIF5, (b) FXR1 and (c) ECHS1 with or without erastin induction. (d) Nuclear/cytoplasmic fluorescence intensity ratio of EIF5, FHFXR1 and ECHS1 in (a) to (c). Values are the mean \pm S.D. of at least 10 cells per condition.

Besides, thank you for pointing out Figure 2 and we removed it to the supplementary information. We also rearranged the images in the manuscript to make them more informative. “ μm ” was replaced by “ μM ” in page 8 and we are sorry for this typo.

Reviewer #2 (Remarks to the Author):

In this paper, Sun et al. present a comprehensive chemoproteomic profiling of nuclear and cytoplasmic RNA-binding proteins (RBPs) using organelle-targeting furocoumarin-biotin probes. This method represents a significant technical advancement compared to previous approaches for mapping nuclear RBPs. Furthermore, the authors successfully apply this method to investigate the subcellular-specific RBPs involved in ferroptosis, providing valuable insights into the regulation of RNA-protein interactions during this cellular process. The study also uncovers potential candidates for nucleoplasmic translocation, which are further confirmed through immunofluorescence imaging.

Overall, this novel method captures broad interest within the scientific community. The

datasets containing subcellular RBPs, particularly those related to ferroptosis, are of immense value and contribute to our understanding of protein trafficking during this cellular phenomenon. Therefore, I highly recommend this publication for consideration in Nature Communications, with only minor revisions.

We thank the reviewer for the positive evaluation of our manuscript. We made some changes as suggested. The related figures and descriptions were added and colored in red in the manuscript and supporting information.

1. The label "RNase A washing" in Fig. 1d is misleading since RNase A washing is used as a control in Fig. 1e. It would be more appropriate to label it as "RNase A elution" in Fig. 1d.

Thanks for the reviewer's suggestion. We changed the description in Fig. 1d from "RNase A washing" to "RNase A elution" as suggested.

2. As mentioned in the original paper, the authors should change "APEX-OAPS" to "APEX-PS" to maintain consistency.

Thanks for the reviewer's reminding. We changed "APEX-OAPS" to "APEX-PS" to maintain consistency.

3. In order to better understand the advantages of this subcellular-specific RNA labeling method, I suggest that the authors further compare it with previous datasets obtained from serIC, RBR-ID, and APEX-PS. This would provide a more comprehensive evaluation of the method's strengths in subcellular-specific RBP profiling.

Thanks for the reviewer's suggestion. RBR-ID was applied to mouse embryonic stem cells (ESCs), while the other three methods were all applied to human cell lines. Furthermore, nucleus was the only common organelle that was studied by all the latter three methods. Therefore, the nuclear RBPs obtained by serIC, APEX-PS and our probe was compared in Figure R11. Our nuclear probes enriched the most nuclear RBPs among the three methods and covered 42.4% RBPs that obtained by serIC and APEX-PS. The moderately overlapped nuclear RBPs identified by the three methods maybe attributed to two possible reasons. First, different cell lines were used, K562 cells for serIC, HEK293T cells for APEX-PS and HeLa cells for our probes. Second, the enrichment mechanisms are different in the three methods, which may introduce certain bias in RBP enrichment (serIC relies on oligo(dT)-polyadenylated RNA interaction, APEX-PS using organic-aqueous phase separation and our probe is based on the covalent binding between furocoumarin and uracil).

Figure R11. Overlap of the nuclear RBPs identified by serIC, APEX-PS and by nuclear probe in this work.

Reviewer #3 (Remarks to the Author):

The manuscript by Haofan Sun and colleagues presents a novel method utilizing furocoumarin-containing probes to selectively label and isolate nuclear or cytoplasmic RNAs. Through in-cellulo UV-crosslinking of proteins and RNA, the researchers isolated ribonucleoprotein complexes from the nucleus and cytoplasm. The study identified a significant number of nuclear RBPs (1221) and cytoplasmic RBPs (1333), allegedly surpassing previous techniques. Moreover, the authors investigated changes in the nuclear and cytosolic RBPomes following treatment with three ferroptosis inducers. They identified numerous responsive proteins including 59 RBPs defined as "nucleoplasmic translocation candidates". Overall, this approach offers an intriguing opportunity for globally characterizing RBPs with subcellular resolution across biological contexts.

We thank the reviewer for the positive evaluation of our manuscript.

However, there are a few areas that require attention. Firstly, the lack of commercially available probes could limit the wider applicability of the method. Secondly, the authors should include instrumental controls to assess the technique's specificity. Lastly, it would greatly enhance the study's value to explore the biological significance of observed responses or investigate certain aspects of the underlying mechanisms. Addressing these points would further strengthen the overall findings and implications of the research.

We appreciated the reviewer's recognition of the potential of our method and the valuable suggestion on further exploring its application in discovering biological insights of ferroptosis. The synthesis route was described in detail (please refer to

answer to Q2) to facilitate adoption of BLTF and BETF probes by other labs for large-scale study of nucleus and cytoplasm RBPs.

We agree with the reviewer that strict quality control of the mass spectrometer is necessary to obtain reliable results. The Orbitrap Fusion™ Tribrid™ mass spectrometer we used was routinely monitored using tryptic digested HEK293T whole cell lysate. A minimum of 4,000 proteins, 20,000 peptides and 30,000 PSMs was required to demonstrate good instrument condition. Table R2 showed all the quality control data before conducting RBPs analysis.

Table R2. Mass spectrometry quality control by tryptic digested HEK293T whole cell lysate.

Running Date	Proteins	Peptides	PSM
20210331	4870	28087	37365
20210716	4173	23080	31445
20210720	4197	23376	31068
20210805	4402	23284	31394
20211025	4527	27527	37275
20211027	4887	31903	42488
20221123	4514	30388	39858
20221125	4639	31307	40962
20221229	4069	22774	32032
20230105	4338	26429	35452
20230811	4564	29794	38513
20230828	4852	29458	38280
20230927	4595	24620	36017

Major comments:

Q1 To distinguish genuine RNA-binding proteins from proteins that may associate non-specifically (e.g. independent of UV) or indirectly (e.g. through interaction with RBPs) with RNA, it is crucial to include a non-UV control in the experimental design. This control would involve treating the samples identically but without UV crosslinking at 254 nm (the irradiation at 365 nm would still be performed). By comparing the results obtained from the UV-crosslinked samples to those of the non-UV controls (generated in parallel), the specificity of the technique could be assessed in a way that the current control (RNase treatment before the washing step) does not allow. This control should be performed at least with the most effective nuclear and cytosolic probes.

Thanks for the reviewer's suggestion. Although the strong washing condition using "SDS and urea" should efficiently remove the non-RBPs that bound with RNA/RBP/beads, we agree with the reviewer that proper and stringent controls are critical for obtaining reliable RBPs. Every sample treatment step may introduce non-specifically or indirectly bound non-RBPs. Therefore, all the sample processing steps that the experiment group experiences need to be included in the control, so as to

comprehensively exclude the non-RBPs in the experiment groups via quantitative proteomic comparison. In our design, the “RNase washing” allows “identical” sample processing steps in the control group as in the experiment group (Figure R12 below and Figure 1e in the manuscript) to include both UV and non-UV induced non-RBPs (introduced during sample incubation and by non-specific binding with RNA/RBP/beads, etc.). Having said so, we followed the reviewer’s suggestion and compared the proteins identified in different controls using the BLTF and BETF probes in Figure R13 and R14. For both the nuclear and cytoplasmic probes, the identified non-specific proteins in the RNase washing control not only covered almost all the proteins obtained by the non-UV control, but also provided around 70% more non-specific proteins, indicating it’s a more stringent control. Furthermore, abundance comparison between the proteins identified in both of the two types of controls showed a similar trend. The RNase washing control exhibited higher abundance of the non-specific proteins and therefore it can more efficiently screen out the non-RBPs in the experiment group by quantitative comparison.

Figure R12. Experimental design of the quantitative differential proteomic comparison between the experimental group and control group for RBP identification.

Figure R13. Overlap of the non-specific proteins identified from the RNase A washing

control and non-UV control using (a) BLTF and (b) BETF probes.

Figure R14. Boxplots showing abundance of the non-specific proteins identified in the RNase A washing and non-UV controls using (a) BLTF and (b) BETF probes.

Q2 The described protocol is likely be of interest to laboratories with expertise in RNA/RBP biology but not necessarily with a strong background in chemistry. It is advisable for the authors to provide a detailed step-by-step protocol that includes comprehensive information to facilitate the procedure for non-experts. This should include specific instructions on the required equipment, reaction conditions, recommended reagent brands and their purity, as well as any additional comments and quality controls necessary for successful probe synthesis.

Thanks for the reviewer's suggestion. We rewrote the protocol for probe synthesis in a step-by-step manner so that other researchers can reproduce the probes easily.

Synthesis of N-(14-(2,5-dioxo-2,5-dihydro-1H-pyrrol-1-yl)-3,6,9,12-tetraoxatetradecyl)-4-((7-oxo-7H-furo[3,2-g]chromen-9-yl)oxy)butanamide (1)

1. A solution of 2,5-dioxopyrrolidin-1-yl 4-((7-oxo-7H-furo[3,2-g]chromen-9-yl)oxy) butanoate (Thermo, 19.3 mg, 0.05 mmol) in tetrahydrofuran solution (Sigma, $\geq 99.0\%$, 1 mL) was added dropwise into a 10 mL conical flask with saturated sodium bicarbonate solution (Sigma, $\geq 99.7\%$, 2 mL) of 1-(14-amino-3,6,9,12-tetraoxatetradecyl)-1H-pyrrole-2,5-dione (ToYongBio, 31.6 mg, 0.1 mmol) and stirred overnight at room temperature by a magnetic stirrer.
Caution: Thin-layer chromatography (TLC) was applied by glass support silica gel 60 matrix (Merck) to monitor products (DCM/MeOH, 15/1, v/v as eluent).
2. The reaction mixture was transferred to a 15 mL tube. DCM (J&K Scientific, $\geq 99.5\%$ 10 mL) was added to the reaction mixture to extract the products. Repeat the DCM extraction step 3 times.
3. The organic phase was combined and dried over anhydrous sodium sulfate (Sigma, $\geq 99.0\%$, 500 mg), then evaporated to dryness. The crude product was purified by silica column chromatography (DCM/MeOH, 25/1, v/v) to give the compound 1.

Synthesis of Biotin-C(Furocoumarin)YGRKKRRQRRR (Biotin-TAT-Furocoumarin, BTF)

1. A solution of compound 1 (11.7 mg, 0.02 mmol) in aqueous acetonitrile solution (ACN/H₂O = 3:1, v/v, 1 mL) was added into Biotin-C-TAT (SYNPEPTIDE, 18.9 mg, 0.01 mmol). The resulting solution was stirred overnight at room temperature by a magnetic stirrer.

Caution: TLC was applied to monitor the products (DCM/MeOH, 10/1, v/v as eluent).

2. The solvent was removed under reduced pressure and the residue was purified by silica column chromatography (DCM/MeOH, 10/1, v/v) to give the compound BTF.

Synthesis of Biotin-C(Furocoumarin)PKKKRKV (Biotin-NLS-Furocoumarin, BLF)

1. A solution of compound 1 (11.7 mg, 0.02 mmol) in aqueous acetonitrile solution (ACN/H₂O = 3:1, v/v, 1 mL) was added into Biotin-C-NLS (SYNPEPTIDE, 12.1 mg, 0.01 mmol). The resulting solution was stirred overnight at room temperature by a magnetic stirrer.

Caution: TLC was applied to monitor the products (DCM/MeOH, 10/1, v/v as eluent).

2. The solvent was removed under reduced pressure and the residue was purified by silica column chromatography (DCM/MeOH, 10/1, v/v) to give the compound BLF.

Synthesis of Biotin-C(Furocoumarin)PKKKRKVYGRKKRRQRRR (Biotin-NLS-TAT-Furocoumarin, BLTF)

1. A solution of compound 1 (11.7 mg, 0.02 mmol) in aqueous acetonitrile solution (ACN/H₂O = 3:1, v/v, 1 mL) was added into Biotin-C-NLS-TAT (SYNPEPTIDE, 27.5 mg, 0.01 mmol). The resulting solution was stirred overnight at room temperature by a magnetic stirrer.

Caution: TLC was applied to monitor the products (DCM/MeOH, 10/1, v/v as eluent).

2. The solvent was removed under reduced pressure and the residue was purified by silica column chromatography (DCM/MeOH, 10/1, v/v) to give the compound BLTF.

Synthesis of Biotin-C(Furocoumarin)PKKKRKVPKKKRKVPKKKRKV (Biotin-NLS-NLS-NLS-Furocoumarin, BL3F)

1. A solution of compound 1 (11.7 mg, 0.02 mmol) in aqueous acetonitrile solution (ACN/H₂O = 3:1, v/v, 1 mL) was added into Biotin-C-NLS-TAT (SYNPEPTIDE, 29.4 mg, 0.01 mmol). The resulting solution was stirred overnight at room temperature by a magnetic stirrer.

Caution: TLC was applied to monitor the products (DCM/MeOH, 10/1, v/v as eluent).

2. The solvent was removed under reduced pressure and the residue was purified by silica column chromatography (DCM/MeOH, 10/1, v/v) to give the compound BL3F.

Synthesis of Biotin-C(Furocoumarin)LQLPPLERLTD (Biotin-NES-Furocoumarin, BEF)

1. A solution of compound 1 (11.7 mg, 0.02 mmol) in aqueous acetonitrile solution (ACN/H₂O = 3:1, v/v, 1 mL) was added into Biotin-C-NES (SYNPEPTIDE, 17.4 mg, 0.01 mmol). The resulting solution was stirred overnight at room temperature

by a magnetic stirrer.

Caution: TLC was applied to monitor the products (DCM/MeOH, 10/1, v/v as eluent).

2. The solvent was removed under reduced pressure and the residue was purified by silica column chromatography (DCM/MeOH, 10/1, v/v) to give the compound BEF.

Synthesis of Biotin-C(Furocoumarin)LQLPPLERLTLDYGRKKRRQRRR (Biotin-NES-TAT-Furocoumarin, BETF)

1. A solution of compound 1 (11.7 mg, 0.02 mmol) in aqueous acetonitrile solution (ACN/H₂O = 3:1, v/v, 1 mL) was added into Biotin-C-NES-TAT (SYNPEPTIDE, 32.8 mg, 0.01 mmol). The resulting solution was stirred overnight at room temperature by a magnetic stirrer.

Caution: TLC was applied to monitor the products (DCM/MeOH, 10/1, v/v as eluent).

2. The solvent was removed under reduced pressure and the residue was purified by silica column chromatography (DCM/MeOH, 10/1, v/v) to give the compound BETF.

Synthesis of Biotin-C(Furocoumarin)LQLPPLERLTLDPKKKRKY (Biotin-NES-NLS-Furocoumarin, BELF)

1. A solution of compound 1 (11.7 mg, 0.02 mmol) in aqueous acetonitrile solution (ACN/H₂O = 3:1, v/v, 1 mL) was added into Biotin-C-NES-NLS (SYNPEPTIDE, 26.0 mg, 0.01 mmol). The resulting solution was stirred overnight at room temperature by a magnetic stirrer.

Caution: TLC was applied to monitor the products (DCM/MeOH, 10/1, v/v as eluent).

2. The solvent was removed under reduced pressure and the residue was purified by silica column chromatography (DCM/MeOH, 10/1, v/v) to give the compound BELF.

Synthesis of Biotin-C(Furocoumarin)PKKKRKYLQLPPLERLTLD (Biotin-NLS-NES-Furocoumarin, BLEF)

1. A solution of compound 1 (11.7 mg, 0.02 mmol) in aqueous acetonitrile solution (ACN/H₂O = 3:1, v/v, 1 mL) was added into Biotin-C-NLS-NES (SYNPEPTIDE, 26.0 mg, 0.01 mmol). The resulting solution was stirred overnight at room temperature by a magnetic stirrer.

Caution: TLC was applied to monitor the products (DCM/MeOH, 10/1, v/v as eluent).

2. The solvent was removed under reduced pressure and the residue was purified by silica column chromatography (DCM/MeOH, 10/1, v/v) to give the compound BLEF.

The MS spectra and molecular weight of the synthesized probes shown in Figure S2 and Table S1 (Supporting Information) confirmed successful synthesis of the probes.

The above information was added to the manuscript, page 26.

Q3 The reported translocation of TCA enzymes to the nucleus following treatment with ferroptosis inducers is indeed intriguing. Validating and further investigating this observation would certainly enhance the impact of the study. While conducting an exhaustive exploration of the biological significance may be beyond the scope of the work, there are numerous relevant questions that could be addressed within a reasonable time frame. Please find below some suggestions on that line, that I have grouped into three categories:

i) Characterizing the behaviour of the probes and the overall effects of the employed treatments with ferroptosis inducers in cellular biology: Can the BETF probe penetrate mitochondria? Do the described treatments affect the integrity of mitochondria or the nucleus? It is important to confirm that the treatments do not alter the subcellular localization of the BLTF/BETF probes (Figure 6a and Figure S8, now refers to the Figure 6a and Figure S13).

Thank you for pointing this out. We performed confocal fluorescent imaging analysis to show the localization of the BLTF/BETF probes before and after ferroptosis treatment in Figure R15. BLTF/BETF probes do not alter their originally targeted subcellular localization after ferroptosis treatment. Furthermore, MitoTracker and Hoechst 33342 staining indicated that integrity of mitochondria and nucleus was not affected by the ferroptosis inducers. The above information was added to the manuscript, page 17 and supporting information (Figure S11).

Figure R15. Confocal fluorescent imaging of (a) BLTF and (b) BETF probes with and without ferroptosis inducer treatment. The BLTF and BETF probes were stained with FITC (green) via biotin and streptavidin coupling. Hoechst 33342 (blue) was used as a nuclear marker and Mitotracker (red) was used as a mitochondria maker. Scale bars, 20 μm .

ii) Validating the translocation of TCA proteins to the nucleus through orthogonal assays: This can be assessed through imaging, as shown in Figure 7a-c, and by performing subcellular fractionation followed by western blotting/proteomics.

Thank you for pointing this out. Translocation of TCA proteins to the nucleus was validated using subcellular fractionation followed by proteomics analysis and fluorescent imaging. Subcellular fractionation was performed to obtain nucleus and

cytoplasm using the protocol described by Conrad et al²¹. The purified nucleus and cytoplasm fractions were evaluated by immunoblotting using nuclear markers lamin A/C and histone 3, as well as cytoplasmic marker β -tubulin in Figure R16. The nuclear fraction only showed clear bands of Lamin A/C and histone 3, while the cytoplasmic fraction showed β -tubulin band without contamination of nuclear markers, demonstrating excellent nuclear and cytoplasmic purification results.

Figure R16. Validation of subcellular fractionation by western blotting. Lamin A/C and histone 3 were used as nuclear markers and β -tubulin was used as the cytoplasmic marker.

Next, we performed proteome profiling of the nucleus to investigate the translocation of TCA proteins to the nucleus. As shown in the volcano plot in Figure R17, majority of the nucleus proteins (>90%) did not show obvious abundance change upon inducing ferroptosis. The five TCA proteins that were upregulated in nuclear RBP enrichment after erastin treatment were chosen for further analysis. ACO2 and FH were upregulated, while CS and MDH2 were unchanged after inducing ferroptosis in Figure R17. The remaining one (PDHA1) was not identified in the proteome of nuclear fraction. Consistent results were obtained using immunofluorescence imaging in Figure R18. ACO2 and FH were indeed translocated to the nucleus with 31.1% and 19.2% fluorescence enhancement after erastin treatment, while changes in nucleocytoplasmic distribution of CS, MDH2 and PDHA1 were negligible. We further normalized the quantitative changes of nuclear RBP enrichment by their corresponding nuclear protein abundance variation. All the four TCA related RBPs still displayed upregulated RNA binding after normalization (Figure R19). The above results revealed that both increased protein abundance by translocation to nucleus and enhanced RNA binding account for the elevated nuclear RBP enrichment of ACO2 and FH. While, for CS, MDH2 and PDHA1, the main reason should be increased RNA binding. The above information was added to the manuscript, page 20 and supporting information (Figure S17-19).

Figure R17. Quantitative analysis of nuclear proteins in cells treated with or without erastin.

Volcano plot displaying \log_2 fold change (FC) (x-axis) and $-\log_{10}$ P values (y-axis) of the nuclear proteome of the erastin-treated and nontreated cells. Proteins with significant increase after erastin treatment were depicted in red, and those significantly decreased after erastin treatment were depicted in blue.

Figure R18. Immunofluorescence imaging of TCA proteins with and without erastin induction.

(a)-(e) Immunofluorescence images of (a) ACO2, (b) FH, (c) CS, (d) MDH2 and (e) PDHA1 with and without erastin induction. (f) The nuclear/cytoplasmic fluorescence intensity ratio of ACO2, FH, CS, MDH2 and PDHA1 with and without erastin induction. Values are the mean \pm S.D. of at least 10 cells per condition.

Figure R19. Quantitative analysis of the BLTF probe enriched nuclear RBPs in cells treated with or without erastin after normalization by the corresponding protein abundance variation.

Volcano plot displaying log₂ fold change (FC) (x-axis) and $-\log_{10}$ P values (y-axis) of the nuclear RBPs enriched from the erastin-treated and nontreated cells after normalization by the nuclear proteome. Significantly increased nuclear RBPs after erastin treatment were depicted in red, and significantly decreased ones were depicted in blue.

iii) Investigating mechanistic aspects: Do TCA enzymes bind RNA inside or outside mitochondria? Which pool, inside or outside mitochondria, translocates to the nucleus? Is the translocation linked to changes in posttranslational modifications of the proteins? Inside the nucleus, do the proteins associate with chromatin? What are their protein interactors, and how do they change upon treatment? I understand that identifying the RNA interactors through methods like RIP or CLIP could potentially offer valuable insights. However, I acknowledge that these techniques may present challenges in obtaining clean and meaningful data. Therefore, while it could be advantageous to explore this aspect, I do not consider it a necessary requirement for the study.

Thank you for pointing this out. It is very interesting to find out which pool of RBPs, inside or outside mitochondria translocates to the nucleus. However, our BLTF and BETF probes do not enable specific RNA binding analysis inside or outside mitochondria. As for the relationship between translocation and post-translational modifications, we found six RBPs, including EIF2S2, MARS1, DDX1, PSMA4, Septin

2 and HNRNPA2B1, that translocated from the cytoplasm to the nucleus when their phosphorylation levels were significantly reduced by staurosporine (STS) treatment in Figure R20.

Figure R20. STS treatment induced cytoplasm-to-nuclear translocation of RBPs. (a) Volcano plot displaying \log_2 fold change (x-axis) and $-\log_{10}$ P values (y-axis) of TiO_2 enriched phosphopeptides from STS treated and non-treated cells. Phosphopeptides significantly increased after STS treatment were depicted in red and phosphopeptides significantly decreased after STS treatment were depicted in blue. b-g Confocal fluorescent imaging of (b) EIF2S2, (c) MARS1, (d) DDX1, (e) PSMA4, (f) Septin 2 and (g) HNRNPA2B1 with and without STS treatment. RBPs were stained with FITC (green), and Hoechst 33342 (blue) was used as a nuclear marker. Scale bars, 20 μm . h The nuclear/cytoplasmic fluorescence intensity ratio of a to f. Values are the mean \pm S.D. of at least 10 cells per condition.

We then performed chromatin-associated proteins enrichment (ChEP)^{14,15} before and after erastin treatment. 175 RBPs were found to also interacted with chromatin and were referred to as chromatin associated-RBPs (CA-RBPs). STRING database was used to obtain highly confident interacting proteins (interaction score > 0.7) of the 175 CA-RBPs identified by both ChEP and our BLTF enrichment in Figure R21. The CA-RBPs were represented by nodes, and the interactions between the them were indicated by edges. The protein-protein interaction (PPI) network containing 175 nodes and 364 edges with their interactors were listed in SupplementaryData5. Pathway and GO analysis of the total CA-RBPs showed obviously enriched DNA related functions, including (Epigenetic regulation of) gene expression, Histone deacetylase, Chromatin organization/remodeling, Nucleosome assembly and DNA replication (Figure R22).

However, we also found a significant module in the PPI network (green nodes) displaying enriched RNA related functions, such as mRNA Splicing/Spliceosome, Processing of Capped Intron-Containing Pre-mRNA, Metabolism of RNA, RNA structure unwinding and Response to exogenous dsRNA (Figure R23). Furthermore, GO terms that not related to DNA or RNA were also find (Rhythmic process, SUMOylation and regulation of megakaryocyte differentiation) indicating complex regulating roles of the CA-RBPs.

RNA binding behaviors of the 175 CA-RBPs were investigated upon erastin induction. As shown in the volcano plot in Figure R24, 25 of the CA-RBPs exhibited upregulated RNA-binding, while none of them downregulated. 5 of the RNA-binding upregulated CA-RBPs, including PRDX5, SND1, KDM1A, XRCC6 and CEBPB, were reported to regulate ferroptosis^{4,16-20}. For instance, SND1 promotes the degradation of GPX4 by destabilizing the HSPA5 mRNA and suppressing HSPA5 expression, thus promoting ferroptosis in osteoarthritis chondrocytes. The above results were added in the revised manuscript, page 21.

Figure R21. PPI network of the CA-RBPs. The green nodes represent significant module from the PPI network containing 11 nodes determined by MCODE plugin.

Figure R22. Analysis of the 175 CA-RBPs.
 a Pathway analysis of the proteins.
 b GO biological process analysis of the proteins.

Figure R23. Analysis of the 11 proteins from enriched PPI cluster by MCODE.

- a Pathway analysis of the proteins.
- b GO biological process analysis of the proteins.

Figure R24. Quantitative analysis of the CA-RBPs in cells treated with or without erastin. Volcano plot displaying log₂ fold change (FC) (x-axis) and $-\log_{10}$ P values (y-axis) of the CA-RBPs quantified from the erastin-treated and nontreated cells. CA-RBPs with significant increase after erastin treatment were depicted in red, and significantly decreased after erastin treatment were depicted in blue.

Q4- Another intriguing aspect that could be explored in a similar manner is the potential enhanced interaction of translation-related RBPs with RNAs in the nucleus following treatment with ferroptosis inducers. I suggest to validate this response and the alleged modulation of the nucleoplasmic translocation of these proteins.

Thanks for the reviewer's suggestion. As shown in Figure R17, majority of the nuclear proteins did not change their abundance after erastin induction. For the 24 translation-related RBPs exhibited enhanced interaction with RNAs in nucleus after erastin induced ferroptosis, only EIF5 displayed upregulation in the nuclear proteome profiling dataset. To further validate the nucleoplasmic translocation of the translation-related RBPs, we chose one RBP as the representative for each of the three situations in our study. EIF5 represented upregulation in both RBP enrichment by the nuclear BLTF probe and proteome profiling in the nucleus fraction after erastin treatment. FXR1 represented upregulation only in RBP enrichment, while the protein abundance remained unchanged in proteome profiling. Besides the above two situations, we also chose ECHS1 as a negative control to represent RBPs that were unchanged in both RBP enrichment and proteome profiling. As shown in Figure R25, immunofluorescence imaging displayed consistent results with the proteome profiling data of the nuclear fraction. The results demonstrated that EIF5 translocated to nucleus with 72.7% fluorescence enhancement after erastin induction. FXR1 and ECHS1 showed no abundance change in the nucleus in Figure R25d. Therefore, the increased FXR1 in

RBP enrichment after erastin treatment should be attributed to the enhanced binding with the corresponding RNAs in the nucleus. The above results were added in the revised manuscript, page 20.

Figure R25. Immunofluorescence imaging of translation-related RBPs with or without erastin induction. (a)-(c) Immunofluorescence images of (a) EIF5, (b) FXR1 and (c) ECHS1 with or without erastin induction. (d) Nuclear/cytoplasmic fluorescence intensity ratio of EIF5, FHFXR1 and ECHS1 in (a) to (c). Values are the mean \pm S.D. of at least 10 cells per condition.

Q5- The overall writing style of the paper could be improved.

Thanks for the reviewer's suggestion. We polished the paper using professional English editing service (AJE, code 517F-DCFF-38F1-026D-84C8) to improved its writing style. We hope the revised version may reach the high standard of nature communication.

Minor comments:

Q6- The expression "organelle-specific" in the title gives the false impression that the work covers multiple organelles. I propose to use instead something along the line of: "nuclear- and cytoplasmic-specific RNA Tagging..."

Thanks for the reviewer's suggestion. We changed the title to "Nuclear- and Cytoplasmic-specific RNA Tagging Probes for Spatially Resolved Profiling of RNA Binding Proteome and Investigation upon Ferroptosis Induction".

Q7- "First, the RNA tagging probes were introduced during cell culture..." I find this expression a bit confusing.

We apologize for our confusing writing and edited the manuscript text to make it clearer. The sentence was rewrite to “First, the RNA tagging probes were incubated with living cells in the culture dish to allow the probes to enter the cells and positioned to the nucleus or cytoplasm regions.”

Q8- “Figure 1f and 1g shows that overlapping but distinct RBPs were identified by different nucleus/cytoplasm-targeting probes.” Considering that the probes are fairly similar, why the overlap of captured RBPs (among nuclear probes on one side and among cytosolic probes on the other) is relatively modest? Can the authors discuss that? Moreover, can the authors address limitations and strengths of the described probes/approach in the discussion?

Thank you for pointing this out. Molecular dynamics (MD) simulation was used to explore the structural difference of the RBP enrichment probes (BLTF, BL3F, BLF, BETF, BELF and BLEF) and explaining their difference in RBP enrichment. As shown in Figure R26a, for BLTF, the initial predicted structure of the nuclear targeting part (NLS-TAT peptide) is a helix structure and retains the same after 2 ns of MD simulations. The stable helical structure²² makes the RNA binding part (furocoumarin) of BLTF difficult to fold, therefore leaving enough space to bind to various kinds of RNA. On the contrary, the structure of the nuclear targeting part of BL3F (NLS-NLS-NLS peptide) consists of a shorter Helix structure and beta-sheet (Figure R26b). The conformation of BL3F at 2 ns shows completely folded furocoumarin with hydrogen bonds formed inside. This may cause increased steric hindrance between furocoumarin and RNA²³. Therefore, the inhibited binding between BL3F and RNA may lead to a decreased RBP enrichment. For BLF, its subcellular targeting part (NLS peptide) is a shorter helix-like structure (Figure R26c). This structure makes the NLS peptide easily to interfere with furocoumarin. After 2 ns of MD simulations, the NLS peptide gradually interacts with furocoumarin, causing BLF to fold, and therefore reduces the stability of the subcellular targeting part and the RNA tagging ability of the probe. For the cytoplasmic probes, the (MD) simulations of BETF (Figure R26d) exhibited similar structural features as BLTF. The furocoumarin of BETF adopts an unfold structure by the stable helical structure of the NES-TAT peptide, and therefore results in lower steric hindrance to facilitate its binding to RNA. For BLEF and BELF, the relatively flexible^{24,25} cytoplasmic targeting part (NES-NLS or NLS-NES peptide) results in different extents of folding of the furocoumarin part, and therefore reduces its binding with RNA. To summarize, we attributed the moderate overlap of RBP enrichment by the different nuclear/cytoplasmic probes to their different structure features, which may result in varied tagging efficiencies with RNA.

Figure R26. The initial optimized conformation of the probes and the conformation snapshots after 2ns MD simulations.
a-f (a) BLTF, (b) BL3F, (c) BLF, (d) BETF, (e) BLEF and (f) BELF.

Strengths and Limitations of our approach

By combination of the subcellular targeting peptide and the cycloaddition reaction between furocoumarin and uracil, the subcellular-specific RBP enrichment probes developed in this work achieved 54.4% and 85.7% increase in nuclear and cytoplasmic RBPs identification compared with previously reported methods^{26,27}. Furthermore, the enrichment probes bind to RNAs at living cell level, which is advantageous to obtain more accurate subcellular location of the RBPs and their dynamic changes in biological process than via organelle purification after cells lysis, such as serIC²¹. Having said so, our RBP enrichment probes still have some limitations. First, the enrichment probes are not commercially available yet, so the researchers have to prepare the probes by themselves. We provided detailed synthesis protocol in the answer to Q2 to facilitate adoption of this enrichment method by other labs. Furthermore, currently we can only achieve nuclear and cytoplasmic level RBP enrichment. Other organelles, such as mitochondria, are also doable by this method providing the targeting moieties are available (for example triphenylphosphine for mitochondria), but organelles with higher resolution or without membrane structure may require a completely new probe designing. The limitations were added in the revised manuscript, page 24.

Q9- How were the “experiment” samples (Fig 1e, not treated with RNase prior wash) maintained during the RNase treatment of the control samples? One important aspect to consider is to incubate the “experiment” samples under the same conditions as the control samples, including using the same buffer and maintaining the same temperature (e.g., 37 C for 1 hour). This is essential to ensure that any observed differences in protein elution are due to the digestion of RNA and not influenced by other factors such as temperature-induced effects.

Thank you for pointing this out. We agree with the reviewer that the “experiment” samples should experience exactly the same processing condition as that of the controls, except that no RNase was used. The beads in the experimental group were incubated with 20 μ L PBS at 37 $^{\circ}$ C for 1 h, while the controls were treated with 20 μ L of 0.01 mg/ml RNase A in PBS at 37 $^{\circ}$ C for 1 h. We added a flow chart (Figure R27) to further explain the exact processing procedures and conditions for the “experiment” samples and the controls. The above information was also added to the manuscript in page 10 and supporting information (Figure S5).

Figure R27. Detailed experimental design of the quantitative differential proteomic comparison between the experimental group and control group for RBP identification.

Q10- Regarding the analysis of the nuclear and cytoplasmic RNAs (Figure 4, now refers to the Figure 3):

- Quantify the number of reads mapping to intronic and exonic regions in the nuclear and cytoplasmic RNA pools. Intronic regions should be enriched in the nuclear RNA.
- It might be useful to make use of the RNAseq data to quantify the reads mapping to genes and intergenic regions, in order to assess the level of contamination with genomic DNA in the nuclear and cytosolic pools.

Thanks for the reviewer’s suggestion. As shown in Figure R28, intronic regions are enriched in the nuclear RNA isolated by BLTF compared with the cytoplasmic RNA obtained by BETF. Similar trends were found using density gradient centrifugation as the control. Furthermore, intergenic regions accounts for only about 5% and 1% of reads in the enrichment result of BLTF and BETF, indicating relatively low levels of DNA contamination²⁸⁻³⁰. The above information was added to the manuscript in page

12 and supporting information (Figure S8).

Figure R28. RNA-seq quality control metrics by reads counting.

iii) It would be useful to know which fraction of the RNA isolated from nucleus and cytoplasm corresponds to rRNA. This is not observable in the RNAseq data as ribodepletion was applied, but could be easily estimated by capillary electrophoresis (i.e. bioanalyzer).

Thanks for the reviewer's suggestion. We assessed the isolated RNA samples by Agilent 2100 Bioanalyzer before ribodepletion. Figure R29 (Figure 3e and f in manuscript) shows clear, sharp bands of 18 S rRNA and 28 S rRNA with minimal RNA degradation. The related content has been added to the manuscript, page 13.

Figure R29. Capillary electrophoresis analyzing of the (a) BLTF and (b) BETF enriched RNAs by Agilent 2100 Bioanalyzer.

iv) In Figures 4 e and f (now refers to the Figure 3e and f), the RNAs below 200 nt could not only correspond to tRNAs, but also to other small RNAs such as 5 S rRNA, 5.8 S rRNA, snoRNAs and snRNAs.

Thanks for the reviewer's correction. We removed the tRNA related descriptions in the manuscript.

Q11- It would be useful to include a second volcano plot similar to the one shown in

Figure 5f (now refers to the Figure 4f) but corrected by the background. In other words, the suggested volcano plot would represent the ratio in BLTF vs BETF of the ratios shown in Figure 2 FC (now refers to the Figure S6). This means: \log_2 fold-change(FC-BLTF vs FC-BETF), with FC-BLTF and FC-BETF defined as: \log_2 fold-change(experiment group vs control group). For the reasons explained above, the control groups should be non-irradiated samples, rather than the samples treated with RNase prior the wash.

Thanks for the reviewer's suggestion. We agree with reviewer that the quantitative RBPs comparison of BLTF vs BETF should be corrected by controls to screen the true RBPs from the non-specific proteins. Actually, we processed the enrichment data of BLTF and BETF by the same procedure used in Figure S6 of the manuscript, before plotting the volcano chart of the ratio of BLTF vs BETF. In other words, the dots in Figure R30 below and Figure 4f of the manuscript are all RBPs that pass the stringent filtering criteria (fold change >2 , $P < 0.01$ and $FDR < 1\%$ with two or greater unique peptides in at least two tests) stated in the method part. The experiment VS control volcano plots of BLTF and BETF were shown below (Figure R30). Red dots represented the identified RBPs that passed the screening cut-off and were used in Figure R30 below and Figure 4f of the manuscript. Another advantage of our way of displaying data is that p value is included in Figure 4f to represent significance of the differential RBPs in nucleus and cytoplasm. We added the above explanation in the revised manuscript, page 14.

Figure R30. Scatter plot of proteins enriched by (a) BLTF and (b) BETF displaying the \log_2 fold change (x-axis) and $-\log_{10}$ P values (y-axis) for RBPs identification by quantitative differential comparison between the experiment groups and control groups. Red dots represented the identified RBPs with a stringent screening cut-off.

Q12- When determining the subcellular location for RNA binding of the identified RBPs (Figure 5f, now refers to the Figure 4f), if possible, it would be useful to determine the overall abundance of the proteins in nucleus or cytosol, in order to define how RNA binding activity correlates with protein abundance in each compartment. Likewise, the total proteome analysis following treatment with ferroptosis inducers (Figure S7) would greatly benefit if they would be done independently in nucleus and cytosol.

Thanks for the reviewer's suggestion. It is interesting to find out how RNA binding

quantity (RBQ) correlates with protein abundance in each compartment. Protein abundance in nucleus and cytosol were obtained by density gradient centrifugation using Conrad et al.'s protocol as mentioned above. Figure R31 is the scatter plot showing how the RBQ correlates with the RBP abundance in nucleus and cytosol. Most of the RBPs are located in the first and third quadrants indicating that the subcellular-level RBQ of majority of RBPs is consistent with their abundance distribution in the corresponding organelle. We further analyzed the rest RBPs that fall in the second and fourth quadrants of Figure R31, which have opposite behavior in organelle specific RBQ and RBP abundance distribution. GO analysis of the RBPs with obviously higher abundance but relatively lower RBQ in cytoplasm is shown in Figure R32a. RBPs with GO biological process terms such as 'DNA replication', 'regulation of mRNA stability' and 'DNA replication initiation' are enriched. For RBPs with opposite behavior and enhanced abundance but decreased RBQ in nucleus, GO biological process terms such as 'cytoplasmic translation', 'rRNA processing', and 'translational initiation' are enriched. The subcellular proteome analysis after treatment with ferroptosis inducers was also evaluated in Figure R33. Only very small fractions (2.1-9.3%) of the nuclear/cytoplasmic proteome showed significant up- or downregulation, indicating majority of the variation in RBP enrichment can be attributed to the changes in RBQ. The above results were added in the revised manuscript page 17 and supporting information (Figure S12).

Figure R31. Scatter plot of nucleus to cytoplasm ratio of subcellular abundance of RBPs obtained by proteome profiling VS quantity of RBPs enriched by nuclear and cytoplasm probes. Each dot represents a RBP.

Figure R32. Analysis of the subcellular distribution of RBQ correlates with the RBP abundance in nucleus and cytosol.

a GO biological process of the RBPs with obviously higher abundance but relatively lower RBQ in cytoplasm.

b GO biological process of the RBPs with enhanced abundance but decreased RBQ in nucleus.

Figure R33. Quantitative analysis of nuclear/cytoplasmic proteins in cells treated with or without ferroptosis inducer.

a-c Volcano plot displaying \log_2 fold change (FC) (x-axis) and $-\log_{10}$ P values (y-axis) of the nuclear proteome of the (a) erastin-, (b) RSL3-, (c) DAT-treated and nontreated cells. Proteins with significant increase after treatment were depicted in red, and those significantly decreased after erastin treatment were depicted in blue.

d-f Volcano plot displaying \log_2 fold change (FC) (x-axis) and $-\log_{10}$ P values (y-axis) of the cytoplasmic proteome of the (d) erastin-, (e) RSL3-, (f) DAT-treated and nontreated cells. Proteins with significant increase after treatment were depicted in red, and those significantly decreased after erastin treatment were depicted in blue.

Q13- Figure 7: Show a negative control (a protein whose nuclear/cytoplasmic ratio does not respond to the treatment with the ferroptosis inducers). Furthermore, the cytoplasmic area seems to shrink in the treated cells, is this the case? If so, how does

this affect the analysis and quantification performed?

Thanks for the reviewer's suggestion. RPL7A and RPS27A, were selected as negative controls without obvious change in both subcellular proteome profiling and RNA binding enrichment after erastin treatment in both nucleus and cytoplasm (Table R3). As shown in Figure R34, the immunofluorescence images further confirmed that the nuclear/cytoplasmic ratio of RPL7A and RPS27A was not changed (fold-change < 2) after treatment with erastin. The above results were added in the revised manuscript, page 19 and supporting information (Figure S16).

Table R3. Quantitative analysis of RPS27A and RPL7A identified by subcellular proteome profiling and RNA binding profiling before and after erastin treatment.

Data source	Cellular component	RPL7A (fold change)	RPS27A (fold change)
Subcellular proteome profiling	nucleus	1.96	0.72
	cytoplasm	0.60	1.51
RNA binding enrichment	nucleus	0.97	0.83
	cytoplasm	1.08	1.27

Figure R34. Immunofluorescence images of (a) RPL7A and (b) RPS27A as negative control for RBPs with/without erastin induction.

c The nuclear/cytoplasmic fluorescence intensity ratio of RPL7A and RPS27A in a and b. Values are the mean \pm S.D. of at least 10 cells per condition.

Some of the cytoplasmic region did shrink slightly in the ferroptosis inducer treated

cells. However, we think as long as the expression levels of the cytoplasmic proteins does not change, their fluorescence intensity should not change either. Take the three negative controls in Figure R25c and R34 as examples, the cytoplasm of the erastin treated cells shrank slightly, but the nuclear/cytoplasmic ratio of ECHS1, RPL7A and RPS27A did not change after erastin treatment, which are consistent with the subcellular proteome profiling results.

Q14- Figure S10 (now refers to the Figure S15): Concerning the 59 “nucleoplasmic translocation candidates”, I suggest to add a table with their names and the fold change in nucleus and cytoplasm upon erastin treatment. Likewise, I would add the names of the identified responsive proteins in each of the GO clusters shown.

Thanks for the reviewer’s suggestion. We added a table of the 59 nucleoplasmic translocation candidates with their names and the fold change in nucleus and cytoplasm upon erastin treatment in Table R4 and SupplementaryData 4. We also revised Figure S15 to show the names of the RBPs in each GO clusters (Figure R35 shown below).

Table R4. 59 nucleoplasmic translocation candidates with names and the fold change in nucleus and cytoplasm upon erastin treatment.

Protein ID	Gene Names	nuclear fold change	cytoplasmic fold change
O00170	AIP	4.192276001	-5.567944209
O00303	EIF3F	2.427686056	-3.595432917
O00487	PSMD14	2.688031514	-3.045419057
O43447	PPIH	4.593137741	-4.175024668
O60610	DIAPH1	2.498678207	-4.879781087
O60749	SNX2	3.613052368	-4.795429866
O75439	PMPCB	5.00285848	-5.068110148
O94979	SEC31A	5.951810837	-6.142541885
O95373	IPO7	2.454587936	-5.15279007
P07954	FH	6.212322235	-4.187639872
P08758	ANXA5	5.718394597	-2.753923416
P11310	ACADM	5.336831411	-6.229508718
P22695	UQCRC2	2.777246475	-4.319766363
P27694	RPA1	2.598360697	-4.032466888
P30084	ECHS1	3.275493622	-3.423670451
P31939	ATIC	5.591112773	-3.393173218
P33316	DUT	2.726767222	-3.72061793
P33991	MCM4	3.216480891	-6.296440125
P33993	MCM7	2.710006714	-6.231486638
P37235	HPCAL1	4.576071421	-4.558991114
P38606	ATP6V1A	2.670541763	-5.80614535
P42224	STAT1	4.405959447	-6.345950445
P49736	MCM2	3.392018636	-3.902802149

P52789	HK2	3.595268885	-2.946263631
P60891	PRPS1	5.496825536	-2.505378723
P62633	CNBP	3.606033325	-3.163593928
P63010	AP2B1	2.374523163	-5.315688451
P67870	CSNK2B	3.219921748	-4.719546
P78347	GTF2I	2.833918254	-4.017251968
P78371	CCT2	8.84417216	-2.77579689
Q00796	SORD	5.856503169	-4.476568222
Q02790	FKBP4	3.433543523	-2.461879094
Q06323	PSME1	5.322894414	-2.452273687
Q07960	ARHGAP1	3.735912959	-3.660488764
Q12769	NUP160	4.213575363	-3.248193105
Q13310	PABPC4	7.381341298	-4.638347626
Q13813	SPTAN1	3.107009888	-5.071805954
Q14677	CLINT1	3.816006343	-4.580948512
Q14914	PTGR1	3.132728577	-2.75868543
Q15046	KARS1	5.243430456	-2.943177541
Q15155	NOMO1	2.967114766	-4.309309642
Q16543	CDC37	4.843361537	-3.639350891
Q53EL6	PDCD4	2.97514534	-5.651556015
Q6YN16	HSDL2	4.02485021	-2.823759715
Q8N1G4	LRRC47	5.245452245	-6.149532954
Q8TAT6	NPLOC4	3.145895004	-4.921698252
Q96I24	FUBP3	3.098641713	-3.270256042
Q96QK1	VPS35	4.811002096	-5.907925288
Q99798	ACO2	4.09522438	-6.103830973
Q9NUU7	DDX19A	5.282093048	-3.255360921
Q9UBQ5	EIF3K	5.494370778	-5.570020676
Q9UBQ7	GRHPR	4.651508331	-2.387372971
Q9UHD1	CHORDC1	2.830886841	-2.85699145
Q9UHV9	PFDN2	6.552538554	-6.633529663
Q9UKK9	NUDT5	4.311364492	-5.53613472
Q9Y262	EIF3L	4.517791748	-5.194067001
Q9Y490	TLN1	4.331918716	-2.722389857
Q9Y4L1	HYOU1	4.720127741	-5.732747396
Q9Y5K5	UCHL5	4.766760508	-4.47533226

Figure R35. GO biological process analysis of the erastin-induced nucleoplasmic translocation candidate RBPs.

Reference

1. Gao W, Wang X, Zhou Y, Wang, X, Yu Y. Autophagy, ferroptosis, pyroptosis, and necroptosis in tumor immunotherapy. *Signal Transduc. Tar.* **7**, 196 (2022).
2. Fuchs Y, Steller H. Live to die another way: modes of programmed cell death and the signals emanating from dying cells. *Nat. Rev. Mol. Cell Bio.* **16**, 329-344 (2015).
3. Yang WS, et al. Regulation of ferroptotic cancer cell death by GPX4. *Cell* **156**, 317-331 (2014).
4. Zhang W, et al. RBMS1 regulates lung cancer ferroptosis through translational control of SLC7A11. *J. Clin. Invest.* **131**(2021).
5. Zhang Z, et al. RNA-binding protein ZFP36/TTP protects against ferroptosis by regulating autophagy signaling pathway in hepatic stellate cells. *Autophagy* **16**, 1482-1505 (2020).
6. Lin Z, et al. Hypoxia-induced HIF-1 α /lncRNA-PMAN inhibits ferroptosis by promoting the cytoplasmic translocation of ELAVL1 in peritoneal dissemination from gastric cancer. *Redox Biol.* **52**, 102312 (2022).
7. Xue X, et al. Tumour cells are sensitised to ferroptosis via RB1CC1 - mediated transcriptional reprogramming. *Clin. Transl. Med.* **12**, e747 (2022).
8. Kodiha M, Stochaj U. Nuclear transport: a switch for the oxidative stress—signaling circuit? *J. Sig. Transd.* **2012**(2012).
9. Van Nostrand EL, et al. A large-scale binding and functional map of human RNA-binding proteins. *Nature* **583**, 711-719 (2020).
10. Abildgaard MH, Brynjólfssdóttir SH, Frankel, LB. The autophagy–RNA interplay: degradation and beyond. *Trends Biochem. Sci.* **45**, 845-857 (2020).
11. Anczuków O, et al. The splicing factor SRSF1 regulates apoptosis and proliferation to promote mammary epithelial cell transformation. *Nat. Struct. Mol. Biol.* **19**, 220-228 (2012).
12. Meng L, et al. METTL14 suppresses pyroptosis and diabetic cardiomyopathy by downregulating TINCR lncRNA. *Cell Death Dis.* **13**, 38 (2022).
13. Callow MG, et al. CRISPR whole-genome screening identifies new necroptosis regulators and RIPK1 alternative splicing. *Cell Death Dis.* **9**, 261 (2018).
14. Kustatscher G, Wills KL, Furlan C, Rappsilber J. Chromatin enrichment for proteomics. *Nat. Protoc.* **9**, 2090-2099 (2014).
15. Kustatscher G, et al. Proteomics of a fuzzy organelle: interphase chromatin. *EMBO J.* **33**, 648-664 (2014).
16. Farooqi AA, Kapanova G, Kalmakhanov S, Kussainov AZ, Datkhayeva Z. Regulation of Ferroptosis by non-coding RNAs: Mechanistic insights. *J. Pharmacol. Exp. Ther.* **384**, 20-27 (2023).
17. Qi W, et al. LncRNA GABPB1-AS1 and GABPB1 regulate oxidative stress during erastin-induced ferroptosis in HepG2 hepatocellular carcinoma cells. *Sci. Rep.* **9**, 16185 (2019).
18. Lv M, et al. The RNA-binding protein SND1 promotes the degradation of GPX4 by destabilizing the HSPA5 mRNA and suppressing HSPA5 expression, promoting ferroptosis in osteoarthritis chondrocytes. *Inflamm. Res.* **71**, 461-472

- (2022).
19. Lu C, et al. Aberrant expression of KDM1A inhibits ferroptosis of lung cancer cells through up-regulating c-Myc. *Sci. Rep.* **12**, 19168 (2022).
 20. Zhang S, et al. Role of ferroptosis-related genes in periodontitis based on integrated bioinformatics analysis. *PloS One* **17**, e0271202 (2022).
 21. Conrad T, et al. Serial interactome capture of the human cell nucleus. *Nat. Commun.* **7**, 11212 (2016).
 22. Abrusán G, Marsh JA. Alpha helices are more robust to mutations than beta strands. *PLoS Comput. Biol.* **12**, e1005242 (2016).
 23. Richardson JS, Richardson DC. Natural β -sheet proteins use negative design to avoid edge-to-edge aggregation. *Proc. Natl. Acad. Sci. USA* **99**, 2754-2759 (2002).
 24. Emberly EG, Mukhopadhyay R, Tang C, Wingreen NS. Flexibility of β - sheets: principal component analysis of database protein structures. *Proteins* **55**, 91-98 (2004).
 25. Emberly EG, Mukhopadhyay R, Wingreen NS, Tang C. Flexibility of α -helices: Results of a statistical analysis of database protein structures. *J. Mol. Biol.* **327**, 229-237 (2003).
 26. Qin W, Myers SA, Carey DK, Carr SA, Ting, AY. Spatiotemporally-resolved mapping of RNA binding proteins via functional proximity labeling reveals a mitochondrial mRNA anchor promoting stress recovery. *Nat. Commun.* **12**, 4980 (2021).
 27. Backlund M, et al. Plasticity of nuclear and cytoplasmic stress responses of RNA-binding proteins. *Nucleic Acids Res.* **48**, 4725-4740 (2020).
 28. Bahrami-Samani E, Xing Y. Discovery of allele-specific protein-RNA interactions in human transcriptomes. *Am. J. Hum. Genet.* **104**, 492-502 (2019).
 29. Best MG, In't Veld SG, Sol N, Wurdinger T. RNA sequencing and swarm intelligence-enhanced classification algorithm development for blood-based disease diagnostics using spliced blood platelet RNA. *Nat. Protoc.* **14**, 1206-1234 (2019).
 30. Zhao S, Zhang Y, Gamini R, Zhang B, Von Schack D. Evaluation of two main RNA-seq approaches for gene quantification in clinical RNA sequencing: polyA+ selection versus rRNA depletion. *Sci. Rep.* **8**, 4781 (2018).

REVIEWERS' COMMENTS

Reviewer #1 (Remarks to the Author):

The authors have done a great job in revising the manuscript and the quality and logic have much more improved. I supports its publication. One minor point related to the writing style -- the results section reads more like a combination of figure legends and it is suggested that the authors should describe these exciting results in a more smooth and fluent manner (e.g. to avoid using "Figure XX shows" or "Images in figure XXX shows). Figures and tables should be referenced within parenthesis at the end of each sentences, not but not at the beginning or in the middle.

Reviewer #2 (Remarks to the Author):

The authors have satisfactorily addressed all my concerns.

Reviewer #3 (Remarks to the Author):

The authors have appropriately addressed my comments, and I am pleased to recommend the publication of the manuscript in Nature Communications.

I have a few minor suggestions to enhance the clarity and readability of the manuscript:

1. Can you incorporate a version of Table R4 from the rebuttal letter into the paper, adding additional columns depicting changes in protein abundance in the nucleus and cytoplasm following erastin treatment?
2. I was unable to locate the reference to Supplementary Data 3 in the main text.
3. In the Supplementary Data files, please provide definitions for column names. For example, clarify the meanings of "L," "M," "H," and "S1-3." Specifically, in Supplementary Data 3, explain the presence of "M"

and "L" columns. Additionally, ensure all columns are named. For instance, in Supplementary Data 1, sheets BLTF, BL3F, BETF, BELF, and BLEF contain two unnamed columns.

4. Could you include a column with gene names in Supplementary Data 3 (as seen in Supplementary Data 1)? This addition would enhance the readability of the table for readers.

5. Consider adding a reference to Supplementary Data 4 at the end of the following phrase: 'The 59 RBPs displaying both nuclear upregulation and cytoplasmic downregulation upon erastin treatment were considered as nucleoplasmic translocation candidates (Fig. S14c).

REVIEWERS' COMMENTS

Reviewer #1 (Remarks to the Author):

The authors have done a great job in revising the manuscript and the quality and logic have much more improved. I supports its publication. One minor point related to the writing style -- the results section reads more like a combination of figure legends and it is suggested that the authors should describe these exciting results in a more smooth and fluent manner (e.g. to avoid using "Figure XX shows" or "Images in figure XXX shows). Figures and tables should be referenced within parenthesis at the end of each sentences, not but not at the beginning or in the middle.

We appreciate the reviewer's recognition and revised the writing style in the results part of the manuscript.

Reviewer #2 (Remarks to the Author):

The authors have satisfactorily addressed all my concerns.

We appreciate the reviewer's valuable suggestion and recognition for our manuscript.

Reviewer #3 (Remarks to the Author):

The authors have appropriately addressed my comments, and I am pleased to recommend the publication of the manuscript in Nature Communications.

We appreciate the reviewer's valuable suggestion and recognition for our manuscript.

I have a few minor suggestions to enhance the clarity and readability of the manuscript:

1. Can you incorporate a version of Table R4 from the rebuttal letter into the paper, adding additional columns depicting changes in protein abundance in the nucleus and cytoplasm following erastin treatment?

Thanks for the reviewer's suggestion. We added additional columns depicting changes in protein abundance in the nucleus and cytoplasm following erastin treatment shown in Table R1 and Supplementary Data 4. Since the journal requires 'Large datasets exceeding an A4 page size should be supplied as Supplementary Data files in an extractable format, rather than as Supplementary Tables', we have to keep Table R4 in Supplementary Data 4.

Table R1. 59 nucleoplasmic translocation candidates with names and the fold change in nucleus and cytoplasm with erastin treatment.

Protein ID	Gene Names	nuclear RNA-binding fold change	nuclear protein abundance fold change	cytoplasmic RNA-binding fold change	cytoplasmic protein abundance fold change
------------	------------	---------------------------------	---------------------------------------	-------------------------------------	---

O00170	AIP	4.192276	1.696933	-5.56794	-0.2141
O00303	EIF3F	2.427686	0.747227	-3.59543	1.362216
O00487	PSMD14	2.688032	1.079574	-3.04542	0.237696
O43447	PPIH	4.593138	-0.05837	-4.17502	-2.02703
O60610	DIAPH1	2.498678	5.355691	-4.87978	-0.10936
O60749	SNX2	3.613052	*ND	-4.79543	-0.72531
O75439	PMPCB	5.002858	*ND	-5.06811	0.149391
O94979	SEC31A	5.951811	-2.89509	-6.14254	0.149391
O95373	IPO7	2.454588	-0.22228	-5.15279	-0.15171
P07954	FH	6.212322	4.121128	-4.18764	0.20583
P08758	ANXA5	5.718395	1.091749	-2.75392	-0.13012
P11310	ACADM	5.336831	*ND	-6.22951	0.198858
P22695	UQCRC2	2.777246	1.337255	-4.31977	0.56942
P27694	RPA1	2.598361	0.77799	-4.03247	0.10128
P30084	ECHS1	3.275494	5.497559	-3.42367	0.165625
P31939	ATIC	5.591113	2.645181	-3.39317	0.095654
P33316	DUT	2.726767	4.330004	-3.72062	-0.24281
P33991	MCM4	3.216481	0.433105	-6.29644	0.125627
P33993	MCM7	2.710007	0.258225	-6.23149	-0.07397
P37235	HPCAL1	4.576071	*ND	-4.55899	-0.32907
P38606	ATP6V1A	2.670542	0.118544	-5.80615	0.375035
P42224	STAT1	4.405959	2.058953	-6.34595	0.062694
P49736	MCM2	3.392019	0.376935	-3.9028	0.096989
P52789	HK2	3.595269	2.873854	-2.94626	-0.13314
P60891	PRPS1	5.496826	0.408606	-2.50538	0.22872
P62633	CNBP	3.606033	*ND	-3.16359	0.086997
P63010	AP2B1	2.374523	0.69508	-5.31569	-0.04738
P67870	CSNK2B	3.219922	0.211615	-4.71955	0.033418
P78347	GTF2I	2.833918	0.372348	-4.01725	-0.00964
P78371	CCT2	8.844172	2.397856	-2.7758	0.020878
Q00796	SORD	5.856503	*ND	-4.47657	0.192556
Q02790	FKBP4	3.433544	1.873192	-2.46188	0.33249
Q06323	PSME1	5.322894	*ND	-2.45227	0.122683
Q07960	ARHGAP1	3.735913	*ND	-3.66049	0.92
Q12769	NUP160	4.213575	0.728658	-3.24819	-0.92973
Q13310	PABPC4	7.381341	0.4404	-4.63835	-0.08778
Q13813	SPTAN1	3.10701	0.398462	-5.07181	-0.1966
Q14677	CLINT1	3.816006	1.746957	-4.58095	-0.35863
Q14914	PTGR1	3.132729	*ND	-2.75869	0.183373
Q15046	KARS1	5.24343	0.456182	-2.94318	0.070552
Q15155	NOMO1	2.967115	0.224101	-4.30931	-0.14728
Q16543	CDC37	4.843362	5.931021	-3.63935	-0.43217
Q53EL6	PDCD4	2.975145	*ND	-5.65156	-3.94106

Q6YN16	HSDL2	4.02485	0.007	-2.82376	0.510675
Q8N1G4	LRRC47	5.245452	4.712076	-6.14953	-0.07627
Q8TAT6	NPLOC4	3.145895	*ND	-4.9217	-0.2625
Q96I24	FUBP3	3.098642	0.338148	-3.27026	0.035326
Q96QK1	VPS35	4.811002	2.196589	-5.90793	0.141606
Q99798	ACO2	4.095224	2.295385	-6.10383	0.06193
Q9NUU7	DDX19A	5.282093	0.999702	-3.25536	0.203641
Q9UBQ5	EIF3K	5.494371	0.167948	-5.57002	-0.0462
Q9UBQ7	GRHPR	4.651508	4.944887	-2.38737	0.202548
Q9UHD1	CHORDC1	2.830887	*ND	-2.85699	0.036357
Q9UHV9	PFDN2	6.552539	6.032516	-6.63353	-0.13721
Q9UKK9	NUDT5	4.311364	*ND	-5.53613	-0.57702
Q9Y262	EIF3L	4.517792	0.258162	-5.19407	-0.0044
Q9Y490	TLN1	4.331919	1.850644	-2.72239	0.10941
Q9Y4L1	HYOU1	4.720128	2.394064	-5.73275	0.296888
Q9Y5K5	UCHL5	4.766761	6.575296	-4.47533	-0.12762

*ND: not detected

2. I was unable to locate the reference to Supplementary Data 3 in the main text.

We are sorry for this mistake and the reference to Supplementary Data 3 was added in page 14 of the manuscript.

3. In the Supplementary Data files, please provide definitions for column names. For example, clarify the meanings of "L," "M," "H," and "S1-3." Specifically, in Supplementary Data 3, explain the presence of "M" and "L" columns. Additionally, ensure all columns are named. For instance, in Supplementary Data 1, sheets BLTF, BL3F, BETF, BELF, and BLEF contain two unnamed columns.

We are sorry for this mistake. Definitions and column names were added for Supplementary Data S1-S3. For instance, in Supplementary Data 1 sheet BLF, 'H' was replaced with 'experiment' and 'L' with 'control', while two unnamed columns were filled with '-Log(P-value)' and 'Protein IDs'.

4. Could you include a column with gene names in Supplementary Data 3 (as seen in Supplementary Data 1)? This addition would enhance the readability of the table for readers.

Thanks for the reviewer's suggestion. A column of gene names was added in Supplementary Data 3.

5. Consider adding a reference to Supplementary Data 4 at the end of the following phrase: 'The 59 RBPs displaying both nuclear upregulation and cytoplasmic downregulation upon erastin treatment were considered as nucleoplasmic translocation candidates (Fig. S14c).

Thanks for the reviewer's suggestion. A reference to Supplementary Data 4 was added accordingly.